# MISSSCORE: HIGH-ORDER SCORE ESTIMATION IN THE PRESENCE OF MISSING DATA

## ABSTRACT

The first order derivative (score) of data density, typically estimated via denoising score matching, has emerged as an effective tool for modeling data distribution and generating synthetic data. Extending this concept to higher-order scores could uncover more detailed local information of the data distribution, enabling new applications. However, learning these high-order scores usually requires complete data, which is often unavailable in real-world scenarios such as healthcare and finance due to privacy and cost constraints. In this work, we introduce MissScore, a novel score-based framework for learning high-order scores from observations with missing data. We derive objective functions for estimating high-order scores under different missing data mechanisms and propose a new algorithm to handle missing data effectively. Our empirical results demonstrate that MissScore efficiently and accurately approximates high-order scores with missing data, while enhancing sampling speed and data quality, as validated through several downstream tasks, including data generation and causal discovery.

## 1 INTRODUCTION

The first-order derivative of the log data density, also known as (Stein) score (Liu et al., 2016), plays an important role in various machine learning applications, including image and tabular data synthesis (Song & Ermon, 2019; 2020; Kim et al., 2022), super-resolution (Li et al., 2022), and inverse problems in medical imaging (Song et al., 2021; Chung & Ye, 2022). Denoising Score Matching (DSM) (Vincent, 2011), an efficient method for estimating the score of the data density from samples, has become widely used in training score-based generative models (Ho et al., 2020; Song & Ermon, 2020). Beyond the first-order score, high-order derivatives of the data density, which we refer to as high-order scores, offer more refined local approximations of the data distribution, such as its curvature, and enable new model capabilities. For instance, they can improve the mixing speed of sampling methods (Dalalyan & Karagulyan, 2019; Sabanis & Zhang, 2019; Meng et al., 2021) and provide insights into quantifying the uncertainty in denoising problem Meng et al. (2021). Additionally, Lu et al. (2022) empirically demonstrated that incorporating high-order score matching improves the likelihood of score-based diffusion ordinary differential equations on both synthetic and real data, while maintaining high-quality generation. Furthermore, high-order scores have been utilized in recovering causal structures by bridging the gap between the score of data density and the underlying graph topology (Rolland et al., 2022; Sanchez et al., 2022; Liu et al., 2024).

Despite their promise, learning high-order scores usually requires training the model on complete data (Meng et al., 2021; Lu et al., 2022). However, in many real-world scenarios such as healthcare, finance, and social networks, data often contain missing values due to privacy constraints or high sampling costs (Rubin, 1976; Shpitser, 2016). A straightforward approach to handling missing data is to impute the missing values and train the model on the imputed dataset. Yet, imputation methods can compromise data quality, potentially leading to biased results and significantly degrading performance in downstream tasks (Ouyang et al., 2023). Specifically, imputation methods fail to account for the inherent uncertainty in the missing data, resulting in a distribution over imputed data that does not accurately reflect the true data distribution. Some alternative approaches, such as using generative adversarial networks (GANs) or variational auto-encoders (VAEs) to directly approximate the data generation model from incomplete data (Li et al., 2019; Gain & Shpitser, 2018), require training additional networks, which can be computationally expensive and may also result in model inconsistency. To address this issue, some works propose to explicitly constrain the learning

objective within the model, which can enhance performance (Städler & Bühlmann, 2012; Gao et al., 2022). These studies emphasize the need for alternative unbiased approaches that can directly and more efficiently handle missing data.

Back to score estimation from incomplete data, Ouyang et al. (2023) adopts a similar way by introducing a diffusion-based framework that learns the first-order score directly from incomplete data. In principle, high-order scores could be estimated from a learned first-order score model trained on incomplete data using automatic differentiation. However, this approach becomes computationally impractical for high-dimensional data and large model sizes, particularly when using deep neural networks based models (Meng et al., 2021). Furthermore, automatic differentiation introduces additional estimation errors, as small errors does not always lead to a small estimation error for high-order scores. Moreover, methods based on GANs (Goodfellow et al., 2020) or VAEs (Kingma, 2013) do not inherently capture score information, regardless of whether data is missing or complete. In contrast, score-based models naturally integrate this information (Li et al., 2019; Ho et al., 2020; Gain & Shpitser, 2018). These limitations highlight the need of using score-based models when estimating high-order scores in the presence of missing data.

**Contributions.** In this work, we propose a novel score-based framework, which we call MissScore, for learning high-order scores from incomplete data. We derive objective functions for estimating high-order scores under different missing mechanisms, leveraging DSM to recover the true score function. While our framework can be applied to scores of any order, we focus on second-order scores (i.e., the Hessian of the log density) in our experiments. Our empirical results demonstrate that the proposed models efficiently and accurately approximate high-order scores with missing data. Moreover, we show that our high-order score-based model enhances both sampling speed and data quality in data generation tasks, even with missing data. The quality of these generated samples is further validated across several downstream tasks. Additionally, we introduce a novel causal discovery method for missing data, which scales effectively with the number of variables and samples, and performs competitively with state-of-the-art methods (Gao et al., 2022; Vo et al., 2024) in causal discovery.

The paper is organized as follows: Section 2 introduces the background on high-order DSM and formulates the objective functions for estimating high-order scores under different missing data mechanisms. In Section 3, we detail the proposed training method and evaluate the accuracy of the learned scores. Sections 4 and 5 showcase experimental results on downstream tasks, illustrating the method's effectiveness. Finally, Section 6 concludes the paper. The Appendix includes related work, proofs, and supplementary experimental details.

## 2 ESTIMATING HIGH-ORDER SCORES WITH MISSING DATA

In this section, we provide background on high-order denoising score matching and present our method to deal with missing data.

### 2.1 BACKGROUND ON HIGH-ORDER DENOISING SCORE MATCHING

Consider a data distribution $p_{\text{data}}(\mathbf{x})$ and a model distribution $p(\mathbf{x}; \boldsymbol{\theta})$ over $\mathbb{R}^d$. The score functions of $p_{\text{data}}(\mathbf{x})$ and $p(\mathbf{x}; \boldsymbol{\theta})$ are denoted as $\mathbf{s}_1(\mathbf{x}) = \nabla_{\mathbf{x}} \log p_{\text{data}}(\mathbf{x})$ and $\mathbf{s}_1(\mathbf{x}; \boldsymbol{\theta}) = \nabla_{\mathbf{x}} \log p(\mathbf{x}; \boldsymbol{\theta})$, respectively. In DSM, instead of directly estimating the score function from the original data, the method works by introducing noise from a predefined noise distribution $q_\sigma(\tilde{\mathbf{x}}|\mathbf{x})$ into the data. The objective is then to estimate the score of the perturbed data distribution $q_\sigma(\tilde{\mathbf{x}}) = \int q_\sigma(\tilde{\mathbf{x}}|\mathbf{x}) p_{\text{data}}(\mathbf{x}) d\mathbf{x}$. To achieve so, DSM minimizes the following objective function

$$\frac{1}{2} \mathbb{E}_{p_{\text{data}}(\mathbf{x})} \mathbb{E}_{q_\sigma(\tilde{\mathbf{x}}|\mathbf{x})} \left[ \|\mathbf{s}_1(\tilde{\mathbf{x}}; \boldsymbol{\theta}) - \nabla_{\tilde{\mathbf{x}}} \log q_\sigma(\tilde{\mathbf{x}}|\mathbf{x})\|_2^2 \right]. \quad (1)$$

It has been shown that minimizing Eq. (1) is equivalent to minimizing the score matching loss between $\mathbf{s}_1(\tilde{\mathbf{x}}; \boldsymbol{\theta})$ and $\mathbf{s}_1(\tilde{\mathbf{x}})$ (Vincent, 2011) under certain regularity conditions. When the noise distribution $q_\sigma(\tilde{\mathbf{x}}|\mathbf{x})$ is Gaussian, i.e., $\mathcal{N}(\tilde{\mathbf{x}}|\mathbf{x}, \sigma^2 \mathbf{I})$, the objective simplifies to

$$\mathcal{L}_{\text{DSM}}(\boldsymbol{\theta}) = \frac{1}{2} \mathbb{E}_{p_{\text{data}}(\mathbf{x})} \mathbb{E}_{q_\sigma(\tilde{\mathbf{x}}|\mathbf{x})} \left[ \left\| \mathbf{s}_1(\tilde{\mathbf{x}}; \boldsymbol{\theta}) + \frac{1}{\sigma^2} (\tilde{\mathbf{x}} - \mathbf{x}) \right\|_2^2 \right]. \quad (2)$$

The learned score function implicitly learns how to "denoise" the perturbed data $\tilde{\mathbf{x}}$, guiding it back toward the true data distribution through the optimization of Eq. (2). By focusing on estimating the score of the noise-perturbed distribution $q_\sigma(\tilde{\mathbf{x}})$ instead of the original data distribution $p_{\text{data}}(\mathbf{x})$, DSM offers a more efficient approach to score estimation compared to other techniques (Hyvärinen & Dayan, 2005; Song et al., 2020). When $\sigma$ approaches zero, the perturbed distribution $q_\sigma(\tilde{\mathbf{x}})$ closely approximates $p_{\text{data}}(\mathbf{x})$, meaning that the score estimated by DSM for $q_\sigma(\tilde{\mathbf{x}})$ is nearly identical to that of $p_{\text{data}}(\mathbf{x})$. Meng et al. (2021) provide a derivation of DSM using Tweedie's formula (Efron, 2011), and they generalize this approach to incorporate high-order moments of $\mathbf{x}$ based on $\tilde{\mathbf{x}}$, allowing them to develop an objective function for learning high-order scores.

**Theorem 1** *(Meng et al., 2021)* $\mathbb{E}[\otimes^n \mathbf{x}|\tilde{\mathbf{x}}] = f_n(\tilde{\mathbf{x}}, \mathbf{s}_1, ..., \mathbf{s}_n)$, *where* $\otimes^n \mathbf{x} \in \mathbb{R}^{D^n}$ *denotes $n$-fold tensor multiplications,* $f_n(\tilde{\mathbf{x}}, \mathbf{s}_1(\tilde{\mathbf{x}}), ..., \mathbf{s}_n(\tilde{\mathbf{x}}))$ *is a polynomial of* $\tilde{\mathbf{x}}$, $\mathbf{s}_1(\tilde{\mathbf{x}})$, ..., $\mathbf{s}_n(\tilde{\mathbf{x}})$, *and* $\mathbf{s}_k(\tilde{\mathbf{x}})$ *represents the $k$-th order score of* $q_\sigma(\tilde{\mathbf{x}}) = \int p_{\text{data}}(\mathbf{x}) q_\sigma(\tilde{\mathbf{x}}|\mathbf{x}) d\mathbf{x}$.

Theorem 1 shows that there exists an equality between high-order moments of the posterior distribution of $\mathbf{x}$ given $\tilde{\mathbf{x}}$ and high-order scores with respect to $\tilde{\mathbf{x}}$. Leveraging Theorem 1 and the least squares estimation of $\mathbb{E}[\otimes^k \mathbf{x}|\tilde{\mathbf{x}}]$, the objectives for approximating the $k$-th order scores $\mathbf{s}_k(\tilde{\mathbf{x}})$ can be constructed as follows.

**Theorem 2** *(Meng et al., 2021) Given score functions* $\mathbf{s}_1(\tilde{\mathbf{x}}), \ldots, \mathbf{s}_{k-1}(\tilde{\mathbf{x}})$, *a $k$-th order score model* $\mathbf{s}_k(\tilde{\mathbf{x}}; \boldsymbol{\theta})$ *can be obtained by optimizing the following objective:*

$$\boldsymbol{\theta}^* = \arg\min_{\boldsymbol{\theta}} \mathbb{E}_{p_{\text{data}}(\mathbf{x})} \mathbb{E}_{q_\sigma(\tilde{\mathbf{x}}|\mathbf{x})} \left[ \left\| \otimes^k \mathbf{x} - f_k(\tilde{\mathbf{x}}, \mathbf{s}_1(\tilde{\mathbf{x}}), \ldots, \mathbf{s}_{k-1}(\tilde{\mathbf{x}}), \mathbf{s}_k(\tilde{\mathbf{x}}; \boldsymbol{\theta})) \right\|_2^2 \right].$$

*where* $f_k$ *is a polynomial of* $\{\tilde{\mathbf{x}}, \mathbf{s}_1(\tilde{\mathbf{x}}), \ldots, \mathbf{s}_k(\tilde{\mathbf{x}})\}$ *such that*

$$f_k(\tilde{\mathbf{x}}, \mathbf{s}_1(\tilde{\mathbf{x}}), \ldots, \mathbf{s}_k(\tilde{\mathbf{x}})) = \begin{cases} \tilde{\mathbf{x}} + \sigma^2 \mathbf{s}_1(\tilde{\mathbf{x}}), & \text{if } k = 1, \\ \sigma^2 \frac{\partial}{\partial \tilde{\mathbf{x}}} f_{k-1}(\tilde{\mathbf{x}}, \mathbf{s}_1, \ldots, \mathbf{s}_{k-1}) \\ \quad + \sigma^2 f_{k-1}(\tilde{\mathbf{x}}, \mathbf{s}_1, \ldots, \mathbf{s}_{k-1}) \otimes \left( \mathbf{s}_1(\tilde{\mathbf{x}}) + \frac{\tilde{\mathbf{x}}}{\sigma^2} \right), & \text{if } k \geq 2. \end{cases} \tag{3}$$

*We have* $\mathbf{s}_k(\tilde{\mathbf{x}}; \boldsymbol{\theta}^*) = \mathbf{s}_k(\tilde{\mathbf{x}})$ *for almost all* $\tilde{\mathbf{x}}$.

## 2.2 High-Order Denoising Score Matching with Missing Data

Consider an observation $\mathbf{x} = (x_1, x_2, \ldots, x_d) \in \mathbb{R}^d$, which is sampled from an unknown data distribution $p_{\text{data}}(\mathbf{x})$. Associated with $\mathbf{x}$, there is a binary mask $\mathbf{m} = (m_1, m_2, \ldots, m_d) \in \{0, 1\}^d$ where $m_i = 1$ indicates that $x_i$ is missing, and $m_i = 0$ indicates that $x_i$ is observed. The observed data can be express as $\mathbf{x}_{\text{obs}} = \mathbf{x} \odot (1 - \mathbf{m}) + \texttt{na} \odot \mathbf{m}$, where $\odot$ is the element-wise multiplication and $\texttt{na}$ indicates the missing value. To perturb the original observation $\mathbf{x}$, we adopt a Gaussian mechanism, generating $\tilde{\mathbf{x}}$ from the conditional distribution $\tilde{\mathbf{x}}|\mathbf{x} \sim \mathcal{N}(\mathbf{x}, \sigma^2 \mathbf{I}_d)$, where $\sigma$ is a pre-specified constant. Therefore, the conditional density function of $\tilde{\mathbf{x}}$ given $\mathbf{x}$ is defined as $q_\sigma(\tilde{\mathbf{x}}|\mathbf{x}) := (2\pi\sigma^2)^{-\frac{d}{2}} \exp\{-\frac{(\tilde{\mathbf{x}}-\mathbf{x})^\top(\tilde{\mathbf{x}}-\mathbf{x})}{2\sigma^2}\}$.

The missing mechanisms can be categorized based on the relationships between the mask $\mathbf{m}$ and the complete data $\mathbf{x}$ as follows Rubin (1976) and the data generation process for each mechanism is detailed in Appendix D:

- Missing Completely at Random: mask $\mathbf{m}$ is independent with the completed data $\mathbf{x}$.
- Missing at Random: mask $\mathbf{m}$ only depends on the observed value $\mathbf{x}_{\text{obs}}$.
- Missing Not at Random: $\mathbf{m}$ depends on the observed value $\mathbf{x}_{\text{obs}}$ and the missing value.

In the following section, we derive the objective functions for training the high-order DSM model, in the presence of missing data under the M(C)AR mechanisms.

### 2.2.1 Missing completely at random (MCAR)

In this section, we start by exploring the first- and second-order scores, then extend the approach to scores of any order under the MCAR assumption. The following theorem presents our first theoretical result, showing that DSM with a missing mask can recover the oracle score, which is the gradient of $\log p_{\text{data}}(\mathbf{x})$ with respect to $\mathbf{x}$.

**Theorem 3** *If the missing mechanism of $\mathbf{x}$ is MCAR, with the missing probability of every element lying between 0 and 1, i.e., $p(m_i = 1) \in [0, 1)$ for all $i \in \{1, 2 \ldots, d\}$. We denote the objective $\mathcal{J}_{\text{DSM}}(\boldsymbol{\theta}) = \mathbb{E}_{\mathbf{x},\mathbf{m}}\mathbb{E}_{\tilde{\mathbf{x}}|\mathbf{x},\mathbf{m}}\left[\left\|\left\{\mathbf{s}_1(\tilde{\mathbf{x}}; \boldsymbol{\theta}) + \frac{1}{\sigma^2}(\tilde{\mathbf{x}} - \mathbf{x})\right\} \odot (\mathbf{1} - \mathbf{m})\right\|_2^2\right]$. Then we have,*

$$\arg\min_{\boldsymbol{\theta}} \mathcal{J}_{\text{DSM}}(\boldsymbol{\theta}) = \arg\min_{\boldsymbol{\theta}} \mathbb{E}_{\mathbf{x}}\mathbb{E}_{\tilde{\mathbf{x}}|\mathbf{x}}\left[\left\|\{\mathbf{s}_1(\tilde{\mathbf{x}}; \boldsymbol{\theta}) - \mathbf{s}_1(\tilde{\mathbf{x}})\} \odot \sqrt{\mathbb{P}(\mathbf{m} = 0)}\right\|_2^2\right].$$

*Furthermore, if there exist unique $\boldsymbol{\theta}^*$ such that $\mathbf{s}_1(\tilde{\mathbf{x}}) = \mathbf{s}_1(\tilde{\mathbf{x}}; \boldsymbol{\theta}^*)$, then $\boldsymbol{\theta}^* = \arg\min_{\boldsymbol{\theta}} \mathcal{J}_{\text{DSM}}(\boldsymbol{\theta})$.*

Since $\mathbb{P}(\mathbf{m} = 0) > 0$, we can conclude that the global optimal of DSM with missing mask coincides with the orcale score. The proof is provided in Appendix C.1. For the second-order score model, building upon Theorem 2, the second-order DSM with missing mask can learn the oracle second-order score, which is the Hessian of $\log p_{\text{data}}(\mathbf{x})$ with respect to $\mathbf{x}$.

**Theorem 4** *Suppose the first-order score $\mathbf{s}_1(\tilde{\mathbf{x}})$ is given, and the missing mechanism of $\mathbf{x}$ is MCAR, with the missing probability of every element lying between 0 and 1, i.e., $p(m_i = 1) \in [0, 1)$ for all $i \in \{1, 2 \ldots, d\}$. We denote the objective*

$$\mathcal{J}_{\text{D}_2\text{SM}}(\boldsymbol{\theta}) = \mathbb{E}_{\tilde{\mathbf{x}},\mathbf{x},\mathbf{m}}\left[\left\|\left\{\mathbf{s}_2(\tilde{\mathbf{x}}; \boldsymbol{\theta}) + \mathbf{s}_1(\tilde{\mathbf{x}})\mathbf{s}_1^\top(\tilde{\mathbf{x}}) + \frac{\mathbf{I} - \mathbf{z}\mathbf{z}^\top}{\sigma^2}\right\} \odot \left\{(\mathbf{1} - \mathbf{m})(\mathbf{1} - \mathbf{m})^\top\right\}\right\|_2^2\right],$$

*where $\mathbf{z} = \frac{\tilde{\mathbf{x}} - \mathbf{x}}{\sigma}$. Then we have,*

$$\arg\min_{\boldsymbol{\theta}} \mathcal{J}_{\text{D}_2\text{SM}}(\boldsymbol{\theta}) = \arg\min_{\boldsymbol{\theta}} \mathbb{E}_{\mathbf{x}}\mathbb{E}_{\tilde{\mathbf{x}}|\mathbf{x}}\left[\left\|\{\mathbf{s}_2(\tilde{\mathbf{x}}; \boldsymbol{\theta}) - \mathbf{s}_2(\tilde{\mathbf{x}})\} \odot \sqrt{\mathbb{P}\left[(\mathbf{1} - \mathbf{m})(\mathbf{1} - \mathbf{m})^\top = 0\right]}\right\|_2^2\right].$$

*Furthermore, if there exist $\boldsymbol{\theta}^*$ such that $\mathbf{s}_2(\tilde{\mathbf{x}}) = \mathbf{s}_2(\tilde{\mathbf{x}}; \boldsymbol{\theta}^*)$, then $\boldsymbol{\theta}^* = \arg\min_{\boldsymbol{\theta}} \mathcal{J}_{\text{D}_2\text{SM}}(\boldsymbol{\theta})$.*

Since $\mathbb{P}\left[(\mathbf{1} - \mathbf{m})(\mathbf{1} - \mathbf{m})^\top = 0\right] > 0$, we can conclude that the global optimal of second-order DSM with missing mask aligns with the oracle second order score. The proof is provided in Appendix C.2.

### 2.2.2 Missing At Random (MAR)

In this section, we investigate the first- and second-order scores under the MAR assumption. MAR allows missingness to depend on the observed data but not on the unobserved (missing) data. This is more realistic than the more restrictive MCAR assumption, where the probability of missingness is independent of both observed and unobserved data. However, the probability of missing data depends on the observed data, which can introduce bias in estimates if the missing data mechanism is ignored. Thus, we cannot learn the oracle score models through objective functions proposed in Section 2.2.1.

Inverse Probability Weighting (IPW) is essential in the MAR framework because it provides a way to correct for the bias introduced by the missing data mechanism (Wooldridge, 2007; Seaman & White, 2013). IPW achieves this by reweighting the observed data points, compensating for the fact that some observations are more likely to be missing than others. Each observed data point is weighted by the inverse of its probability of being observed (i.e., the probability that it was not missing). This means that observations with a higher chance of being missing receive a higher weight, while those with a lower chance of being missing receive a lower weight. By reweighting in this way, IPW ensures that the estimates reflect what would have been observed if there were no missing data, thus mitigating the bias introduced by the MAR mechanism.

In the following theorem, we present our theoretical result that DSM with missing mask through IPW can learn the oracle score, i.e., the gradient of $\log p_{\text{data}}(\mathbf{x})$ w.r.t. $\mathbf{x}$.

**Theorem 5** *If the missing mechanism of $\mathbf{x}$ is MAR, with the missing probability of every element lying between 0 and 1, i.e., $p(m_i = 1) \in [0, 1)$ for all $i \in \{1, 2 \ldots, d\}$. We denote the objective*

$$\mathcal{J}_{DSM}(\boldsymbol{\theta}) = \mathbb{E}_{\mathbf{x},\mathbf{m}}\mathbb{E}_{\tilde{\mathbf{x}}|\mathbf{x},\mathbf{m}}\left[\left\|\left\{\mathbf{s}_1(\tilde{\mathbf{x}}; \boldsymbol{\theta}) + \frac{1}{\sigma^2}(\tilde{\mathbf{x}} - \mathbf{x})\right\} \odot \left\{\frac{\mathbf{1} - \mathbf{m}}{\sqrt{\mathbb{P}[\mathbf{m} = 0|\mathbf{x} = \mathbf{x}]}}\right\}\right\|_2^2\right].$$

*Then we have,*

$$\arg\min_{\boldsymbol{\theta}} \mathcal{J}_{\text{DSM}}(\boldsymbol{\theta}) = \arg\min_{\boldsymbol{\theta}} \mathbb{E}_{\mathbf{x}}\mathbb{E}_{\tilde{\mathbf{x}}|\mathbf{x}}\left[\|\{\mathbf{s}_1(\tilde{\mathbf{x}};\boldsymbol{\theta}) - \mathbf{s}_1(\tilde{\mathbf{x}})\}\|_2^2\right].$$

*Furthermore, if there exist unique $\boldsymbol{\theta}^*$ such that $\mathbf{s}_1(\tilde{\mathbf{x}}) = \mathbf{s}_1(\tilde{\mathbf{x}};\boldsymbol{\theta}^*)$, then $\boldsymbol{\theta}^* = \arg\min_{\boldsymbol{\theta}} \mathcal{J}_{\text{DSM}}(\boldsymbol{\theta})$.*

We can conclude that the global optimal of DSM with missing mask coincides with the oracle score under MAR. The proof is provided in Appendix C.4. Building on Theorem 2, the second-order score model using DSM with a missing mask under MAR can learn the oracle second-order score, which is the Hessian of $\log p_{\text{data}}(\mathbf{x})$ with respect to $\mathbf{x}$.

**Theorem 6** *Suppose the first-order score $\mathbf{s}_1(\tilde{\mathbf{x}})$ is given, if the missing mechanism of $\mathbf{x}$ is MAR, and the missing probability of every element lies between 0 and 1, which is $p(m_i m_j = 0) \in [0, 1)$ for all $i, j \in \{1, 2 \ldots, d\}$, we denote the objective*

$$\mathcal{J}_{\text{D}_2\text{SM}}(\boldsymbol{\theta}) = \mathbb{E}_{\tilde{\mathbf{x}},\mathbf{x},\mathbf{m}}\left[\left\|\left\{\mathbf{s}_2(\tilde{\mathbf{x}};\boldsymbol{\theta}) + \mathbf{s}_1(\tilde{\mathbf{x}})\mathbf{s}_1^\top(\tilde{\mathbf{x}}) + \frac{\mathbf{I} - \mathbf{z}\mathbf{z}^\top}{\sigma^2}\right\} \odot \left\{\frac{(\mathbf{1} - \mathbf{m})(\mathbf{1} - \mathbf{m})^\top}{\sqrt{\mathbb{P}[\mathbf{m}\mathbf{m}^\top = 0|\mathbf{x} = \mathbf{x}]}}\right\}\right\|_2^2\right],$$

*where $\mathbf{z} = \frac{\tilde{\mathbf{x}} - \mathbf{x}}{\sigma}$, then we have,*

$$\arg\min_{\boldsymbol{\theta}} \mathcal{J}_{\text{D}_2\text{SM}}(\boldsymbol{\theta}) = \arg\min_{\boldsymbol{\theta}} \mathbb{E}_{\mathbf{x}}\mathbb{E}_{\tilde{\mathbf{x}}|\mathbf{x}}\left[\|\{\mathbf{s}_2(\tilde{\mathbf{x}};\boldsymbol{\theta}) - \mathbf{s}_2(\tilde{\mathbf{x}})\}\|_2^2\right],$$

*If there exist $\boldsymbol{\theta}^*$ such that $\mathbf{s}_2(\tilde{\mathbf{x}}) = \mathbf{s}_2(\tilde{\mathbf{x}};\boldsymbol{\theta}^*)$, then $\boldsymbol{\theta}^* = \arg\min_{\boldsymbol{\theta}} \mathcal{J}_{\text{D}_2\text{SM}}(\boldsymbol{\theta})$.*

We can conclude that the global optimal of the second-order DSM with missing mask aligns with the oracle Hessian under MAR. The proof is provided in Appendix C.5. We now extend this approach to any desired order in both MCAR and MAR. Theorem 7 indicates that $k$-th order DSM with missing mask can learn the oracle $k$-th order score.

**Theorem 7** *Given score functions $\mathbf{s}_1(\tilde{\mathbf{s}}), \cdots, \mathbf{s}_{k-1}(\tilde{\mathbf{s}})$, and the missing probability of every element lies between 0 and 1, which is $p(m_i = 1) \in [0, 1)$ for all $i \in \{1, 2 \ldots, d\}$. If we correctly model the $k-$th order derivative $\mathbf{s}_k(\tilde{\mathbf{x}})$, there exists $\boldsymbol{\theta}^*$ such that $\mathbf{s}_k(\tilde{\mathbf{x}}, \boldsymbol{\theta}^*) = \mathbf{s}_k(\tilde{\mathbf{x}})$, then*

$$\boldsymbol{\theta}^* = \arg\min_{\boldsymbol{\theta}} \mathbb{E}_{\tilde{\mathbf{x}},\mathbf{x},\mathbf{m}}\left[\|\left\{\otimes^k\mathbf{x} - f_k(\tilde{\mathbf{x}}, \mathbf{s}_1(\tilde{\mathbf{x}}), \ldots, \mathbf{s}_{k-1}(\tilde{\mathbf{x}}), \mathbf{s}_k(\tilde{\mathbf{x}};\boldsymbol{\theta}))\right\} \odot \otimes^k\mathbf{w}\|^2\right],$$

*where $\mathbf{w} = \mathbf{1} - \mathbf{m}$ if the missing mechanism of $\mathbf{x}$ is MCAR, and $\mathbf{w} = \frac{\mathbf{1} - \mathbf{m}}{\mathbb{P}[\mathbf{m}^k = 0|\mathbf{x} = \mathbf{x}]}$ if the missing mechanism of $\mathbf{x}$ is MAR.*

The proof is provided in Appendix C.3. The $k$-th order score model $\mathbf{s}_2(\tilde{\mathbf{x}};\boldsymbol{\theta})$ can be learned from observed data via optimizing the following objective,

$$
\begin{aligned}
&\mathcal{L}_{\text{D}_k\text{SM}}(\boldsymbol{\theta}) \\
&= \mathbb{E}_{\tilde{\mathbf{x}}_{\text{obs}},\mathbf{x}_{\text{obs}},\mathbf{m}}\left[\|\left\{\otimes^k\mathbf{x}_{\text{obs}} - f_k(\tilde{\mathbf{x}}, \mathbf{s}_1(\tilde{\mathbf{x}}_{\text{obs}}), \ldots, \mathbf{s}_{k-1}(\tilde{\mathbf{x}}_{\text{obs}}), \mathbf{s}_k(\tilde{\mathbf{x}}_{\text{obs}};\boldsymbol{\theta}))\right\} \odot \otimes^k\mathbf{w}\|^2\right],
\end{aligned}
\tag{4}
$$

where $f_k$ is a polynomial of $\{\tilde{\mathbf{x}}, \mathbf{s}_1(\tilde{\mathbf{x}}), \ldots, \mathbf{s}_k(\tilde{\mathbf{x}})\}$ is defined in Eq. (3), and $\mathbf{w}$ is defined in Theorem 7.

## 3 TRAINING SCORE MODELS BY HIGH-ORDER DSM WITH MISSING DATA

In this section, we describe the training process for high-order score models in the presence of missing data and evaluate their empirical performance. While our analysis specifically focuses on the first- and second-order scores, the approach can be applied to any order of scores.

Based on Theorem 3 and Theorem 5, the first-order score model $\mathbf{s}_1(\tilde{\mathbf{x}};\boldsymbol{\theta})$ is learned by minimizing the following objective function using the observed data,

$$\mathcal{L}_{\text{DSM}}(\boldsymbol{\theta}) = \mathbb{E}_{\mathbf{x}_{\text{obs}},\mathbf{m}}\mathbb{E}_{\tilde{\mathbf{x}}_{\text{obs}}|\mathbf{x}_{\text{obs}},\mathbf{m}}\left[\left\|\left\{\mathbf{s}_1(\tilde{\mathbf{x}}_{\text{obs}};\boldsymbol{\theta}) + \frac{1}{\sigma^2}(\tilde{\mathbf{x}}_{\text{obs}} - \mathbf{x}_{\text{obs}})\right\} \odot \mathbf{w}_1\right\|_2^2\right],
\tag{5}$$

where $\mathbf{w}_1 = \mathbf{1} - \mathbf{m}$ under MCAR, and $\mathbf{w}_1 = \frac{\mathbf{1} - \mathbf{m}}{\sqrt{\mathbb{P}[\mathbf{m} = 0 | \mathbf{x} = \mathbf{x}_{\text{obs}}]}}$ under MAR.

Similarly, the second-order score model $\mathbf{s}_2(\tilde{\mathbf{x}}; \boldsymbol{\theta})$ is learned using the following objective, as derived in Theorem 4 and Theorem 6,

$$\mathcal{L}_{\text{D}_2\text{SM}}(\boldsymbol{\theta}) = \mathbb{E}_{\tilde{\mathbf{x}}_{\text{obs}}, \mathbf{x}_{\text{obs}}, \mathbf{m}} \left[ \left\| \left\{ \mathbf{s}_2(\tilde{\mathbf{x}}_{\text{obs}}; \boldsymbol{\theta}) + \mathbf{s}_1(\tilde{\mathbf{x}}_{\text{obs}}) \mathbf{s}_1^\top(\tilde{\mathbf{x}}_{\text{obs}}) + \frac{\mathbf{I} - \mathbf{z}_{\text{obs}} \mathbf{z}_{\text{obs}}^\top}{\sigma^2} \right\} \odot \mathbf{w}_2 \right\|_2^2 \right], \quad (6)$$

where $\mathbf{z}_{\text{obs}} = (\tilde{\mathbf{x}} - \mathbf{x})/\sigma$. Under MCAR, $\mathbf{w}_2 = (\mathbf{1} - \mathbf{m})(\mathbf{1} - \mathbf{m})^\top$, and under MAR, $\mathbf{w}_2 = \frac{(\mathbf{1} - \mathbf{m})(\mathbf{1} - \mathbf{m})^\top}{\sqrt{\mathbb{P}[\mathbf{m}\mathbf{m}^\top = 0 | \mathbf{x} = \mathbf{x}]}}$.

However, training the second-order score model, $\mathbf{s}_2(\tilde{\mathbf{x}}; \boldsymbol{\theta})$, requires knowledge of the first-order score $\mathbf{s}_1(\tilde{\mathbf{x}})$. Therefore, we adopt a multi-task objective to train both $\mathbf{s}_1(\tilde{\mathbf{x}}; \boldsymbol{\theta})$ and $\mathbf{s}_2(\tilde{\mathbf{x}}; \boldsymbol{\theta})$ simultaneously,

$$\mathcal{L}_{\text{joint}}(\boldsymbol{\theta}) = \mathcal{L}_{\text{DSM}}(\boldsymbol{\theta}) + \omega \mathcal{L}_{\text{D}_2\text{SM}}(\boldsymbol{\theta}), \quad (7)$$

where $\omega \in \mathbb{R}^+$ is a tunable coefficient. $\mathcal{L}_{\text{DSM}}(\boldsymbol{\theta})$ and $\mathcal{L}_{\text{D}_2\text{SM}}(\boldsymbol{\theta})$ correspond to Eq. (5) and Eq. (6). $\mathbb{P}[\mathbf{m} = 0 | \mathbf{x} = \mathbf{x}_{\text{obs}}]$ and $\mathbb{P}[\mathbf{m}\mathbf{m}^\top = 0 | \mathbf{x} = \mathbf{x}_{\text{obs}}]$ in MAR are estimated using logistic regression models, where the response variable indicates whether the data is missing or observed, and the predictors are the observed variables. In the experiments, missing values are handled by replacing them with 0 for continuous variables and creating a new category for discrete variables. One-hot encoding is then applied to discrete variables. Element-wise multiplication with the mask naturally mitigates the impact of replacing missing values with zeros when computing the objective. The algorithm is provided in Appendix D.

### 3.1 IMPROVING STABILITY WITH VARIANCE REDUCTION

It is important to note that, in order to match the score of the true distribution $p_{\text{data}}(\mathbf{x})$, $\sigma$ needs to be close to zero for both DSM and D$_2$SM, so that $q_\sigma(\tilde{\mathbf{x}})$ closely approximates $p_{\text{data}}(\mathbf{x})$. However, training score models using denoising methods can suffer from high variance when $\sigma$ approaches zero. This challenge motivates the use of variance reduction techniques. Building on existing variance reduction methods for DSM (Song & Kingma, 2021; Meng et al., 2021), we propose tailored variance reduction techniques specifically for training DSM with missing data, as follows

$$\mathcal{L}_{\text{DSM-VR}}(\boldsymbol{\theta}) = \mathcal{L}_{\text{DSM}}(\boldsymbol{\theta}) - \mathbb{E}_{\mathbf{x}_{\text{obs}}, \mathbf{m}} \mathbb{E}_{\mathbf{z} \sim \mathcal{N}(\mathbf{0}, \mathbf{I})} \left[ \left( \frac{2}{\sigma} \mathbf{s}(\mathbf{x}_{\text{obs}}; \boldsymbol{\theta})^\top \mathbf{z} \right) \odot \mathbf{g}_1(\mathbf{x}_{\text{obs}}, \mathbf{m}) + \frac{\| \mathbf{z} \odot \mathbf{g}_1(\mathbf{x}_{\text{obs}}, \mathbf{m}) \|^2}{\sigma^2} \right], \quad (8)$$

where $\mathbf{g}_1(\mathbf{x}_{\text{obs}}, \mathbf{m}) = \mathbf{1} - \mathbf{m}$ under MCAR, and $\mathbf{g}_1(\mathbf{x}_{\text{obs}}, \mathbf{m}) = \frac{\mathbf{1} - \mathbf{m}}{\sqrt{\mathbb{P}[\mathbf{m} = 0 | \mathbf{x} = \mathbf{x}_{\text{obs}}]}}$ under MAR.

For the second-order model with missing data, we implement a variance reduction (VR) technique using antithetic sampling (James, 1985; Meng et al., 2021), which involves utilizing two negatively correlated sample vectors centered around $\mathbf{x}$. The objective function is then formulated as

$$\mathcal{L}_{\text{D}_2\text{SM-VR}}(\boldsymbol{\theta}) = \mathbb{E}_{\mathbf{x}_{\text{obs}}, \mathbf{m}} \mathbb{E}_{\mathbf{z} \sim \mathcal{N}(\mathbf{0}, \mathbf{I})} \left[ \left\{ \boldsymbol{\psi}(\tilde{\mathbf{x}}_{\text{obs}}^+)^2 + \boldsymbol{\psi}(\tilde{\mathbf{x}}_{\text{obs}}^-)^2 + 2 \frac{\mathbf{I} - \mathbf{z}\mathbf{z}^\top}{\sigma} \odot \boldsymbol{\Psi} \right\} \odot \mathbf{g}_2(\mathbf{x}_{\text{obs}}, \mathbf{m}) \right], \quad (9)$$

where the antithetic samples are defined as $\mathbf{x}_{\text{obs}}^+ = \mathbf{x}_{\text{obs}} + \sigma \mathbf{z}$ and $\mathbf{x}_{\text{obs}}^- = \mathbf{x}_{\text{obs}} - \sigma \mathbf{z}$. Here, $\boldsymbol{\psi} = \mathbf{s}_2 + \mathbf{s}_1 \mathbf{s}_1^\top$, and $\boldsymbol{\Psi} = (\boldsymbol{\psi}(\tilde{\mathbf{x}}_{\text{obs}}^+) + \boldsymbol{\psi}(\tilde{\mathbf{x}}_{\text{obs}}^-) - 2\boldsymbol{\psi}(\mathbf{x}_{\text{obs}}))$. Under the MCAR setting, $\mathbf{g}_2(\mathbf{x}_{\text{obs}}, \mathbf{m}) = (\mathbf{1} - \mathbf{m})(\mathbf{1} - \mathbf{m})^\top$, while for MAR, $\mathbf{g}_2(\mathbf{x}_{\text{obs}}, \mathbf{m}) = \frac{(\mathbf{1} - \mathbf{m})(\mathbf{1} - \mathbf{m})^\top}{\sqrt{\mathbb{P}[\mathbf{m}\mathbf{m}^\top = 0 | \mathbf{x} = \mathbf{x}]}}$. The formal analysis of the variance reduction are provided in Appendix C.6 and Appendix C.7.

We perform an empirical analysis to assess the impact of VR on training score models with DSM and D$_2$SM using incomplete data. The full data is generated from a 2-d Gaussian distribution and we simulate the incomplete data under MCAR, training $\mathbf{s}_1(\tilde{\mathbf{x}}_{\text{obs}}; \boldsymbol{\theta})$ and $\mathbf{s}_2(\tilde{\mathbf{x}}_{\text{obs}}; \boldsymbol{\theta})$ using a joint learning objective Eq. (7). Using a sample size of 1000 and a missing ratio of 0.3, In Figure 1, we compare the estimated score and Hessian for the first dimension against the ground truth at noise levels $\sigma = \{0.1, 0.001\}$, both with and without VR. The results indicate that VR is essential for accurate estimation in both DSM and D$_2$SM when $\sigma$ is close to zero, while its importance diminishes at higher values of $\sigma$. As $\sigma$ increases, both methods still achieve reasonable score estimates even without VR. Additionally, when using complete data and varying the missing ratio $\alpha = \{0.1, 0.3, 0.5\}$ with DSM

and VR at $\sigma = 0.001$, we observe that the first- and second-order score estimates remain close to the ground truth. While performance degrades as the proportion of missing data increases, the estimates remain generally accurate.

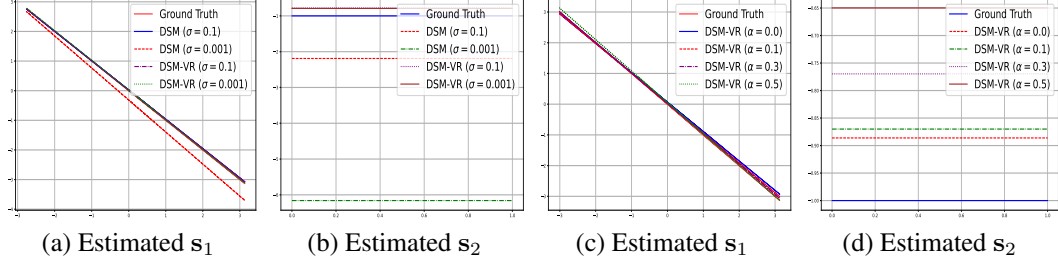

(a) Estimated $\mathbf{s}_1$        (b) Estimated $\mathbf{s}_2$        (c) Estimated $\mathbf{s}_1$        (d) Estimated $\mathbf{s}_2$

Figure 1: Comparison of estimated $\mathbf{s}_1$ and $\mathbf{s}_2$ under different conditions. (a) and (b) show estimates with DSM and $D_2SM$ varying the noise level $\sigma$ with a fixed missing ratio 0.3. (c) and (d) show estimates with DSM and $D_2SM$ varying the missing ratio $\alpha$.

## 3.2 SCALABILITY AND NUMERICAL STABILITY

We show that the proposed method efficiently and accurately estimates second-order scores across different missing ratios, as summarized in Table 1. To achieve this, we generate 10 synthetic datasets with known ground truth, consisting of 100-dimensional correlated multivariate normal distributions that include varying levels of missing data under MCAR mechanism. The covariance matrix for these distributions is constructed using eigenvalues $t \in \{1, 5\}$ to vary degrees of correlation. We evaluate the performance of the estimated $\mathbf{s}_1$ and the diagonal of $\mathbf{s}_2$ by calculating the mean squared error (MSE) between the estimated scores and the ground truth scores derived from the complete data, across various values of $\sigma$. Our results indicate that the jointly optimized $\mathbf{s}_1(\tilde{\mathbf{x}}_{\text{obs}}; \boldsymbol{\theta})$ and $\mathbf{s}_2(\tilde{\mathbf{x}}_{\text{obs}}; \boldsymbol{\theta})$ achieve empirical performance close to the ground truth. As previously mentioned, when $\sigma$ approaches zero, the estimates without VR become unreliable for both the score and Hessian, likely due to convergence issues. Although performance declines with an increase in missing data, the estimates remain reasonable.

Table 1: Mean squared error (MSE) between the estimated first-order and second-order scores and the ground truth is evaluated across 5,000 test samples. We vary the noise scales $\sigma$ and missing ratios $\alpha$, with each configuration tested using 10 random seeds.

| Methods | $\alpha = 0.0$ | | $\alpha = 0.1$ | | $\alpha = 0.3$ | | $\alpha = 0.5$ | |
|---|---|---|---|---|---|---|---|---|
| | $\sigma = 0.1$ | $\sigma = 0.01$ | $\sigma = 0.1$ | $\sigma = 0.01$ | $\sigma = 0.1$ | $\sigma = 0.01$ | $\sigma = 0.1$ | $\sigma = 0.01$ |
| $\mathbf{s}_1$ | $0.28 \pm 0.01$ | $0.42 \pm 0.02$ | $0.29 \pm 0.01$ | $0.42 \pm 0.02$ | $0.32 \pm 0.01$ | $0.44 \pm 0.01$ | $0.37 \pm 0.01$ | $0.44 \pm 0.02$ |
| $\mathbf{s}_1$(VR) | $0.07 \pm 0.00$ | $0.07 \pm 0.00$ | $0.09 \pm 0.00$ | $0.09 \pm 0.00$ | $0.13 \pm 0.00$ | $0.15 \pm 0.01$ | $0.23 \pm 0.00$ | $0.27 \pm 0.01$ |
| $\mathbf{s}_2$ | $0.16 \pm 0.02$ | $15.42 \pm 0.47$ | $0.16 \pm 0.02$ | $27.42 \pm 2.34$ | $0.16 \pm 0.02$ | $29.74 \pm 3.35$ | $0.17 \pm 0.03$ | $26.08 \pm 2.03$ |
| $\mathbf{s}_2$(VR) | $0.04 \pm 0.00$ | $0.05 \pm 0.00$ | $0.04 \pm 0.00$ | $0.04 \pm 0.00$ | $0.04 \pm 0.00$ | $0.05 \pm 0.00$ | $0.05 \pm 0.00$ | $0.06 \pm 0.01$ |

## 4 SAMPLING WITH MISSING DATA VIA SECOND-ORDER SCORE MODELS

In this section, we illustrate how our second-order score model $\mathbf{s}_2(\tilde{\mathbf{x}}; \boldsymbol{\theta})$, trained with missing data, enhances both the speed of sample generation and the quality of the synthetic data. We demonstrate the effectiveness of the proposed model through simulations and real-world datasets containing missing values, comparing its performance against various baseline methods.

**Langevin dynamics.** Langevin dynamics samples data from $p_{\text{data}}(\mathbf{x})$ by utilizing the first-order score function $\mathbf{s}_1(\mathbf{x})$ (Bussi & Parrinello, 2007; Song & Ermon, 2019). Starting with a prior distribution $\pi(\mathbf{x})$, a fixed step size $\epsilon > 0$, and an initial value $\tilde{\mathbf{x}}_0 \sim \pi(\mathbf{x})$, Langevin dynamics iteratively updates the samples as follows:

$$\tilde{\mathbf{x}}_{t+1} = \tilde{\mathbf{x}}_t + \frac{1}{2}\epsilon \mathbf{s}_1(\tilde{\mathbf{x}}_t) + \sqrt{\epsilon}\mathbf{z}_t, \tag{10}$$

where $\mathbf{z}_t \sim \mathcal{N}(\mathbf{0}, \mathbf{I})$ represents Gaussian noise.

**Ozaki Sampling.** Following Meng et al. (2021), Ozaki discretization improves data synthesis by integrating second-order information from $\mathbf{s}_2(\mathbf{x})$ to precondition the sampling process. The updates for Ozaki sampling are performed as follows:

$$\tilde{\mathbf{x}}_t = \tilde{\mathbf{x}}_{t-1} + \mathbf{M}_{t-1}\mathbf{s}_1(\tilde{\mathbf{x}}_{t-1}) + \Sigma_{t-1}^{1/2}\mathbf{z}_t, \tag{11}$$

where $\mathbf{z}_t \sim \mathcal{N}(\mathbf{0}, \mathbf{I})$, $\mathbf{M}_{t-1} = e^{\epsilon\mathbf{s}_2(\tilde{\mathbf{x}}_{t-1})} - \mathbf{I}$, and $\Sigma_{t-1} = (e^{2\epsilon\tilde{\mathbf{x}}_{t-1}} - \mathbf{I})\mathbf{s}_2(\tilde{\mathbf{x}}_{t-1})^{-1}$.

**Illustration.** We use the Swiss-Roll dataset to demonstrate the effectiveness of Ozaki Sampling, focusing on its speed and quality of data generation through second-order information. Both methods employ a step size of $\epsilon = 0.005$ and a missing ratio of $0.5$ under the MCAR missing mechanism. As shown in Figure 2, Ozaki Sampling generates comparable data to Langevin dynamics with fewer iterations, and resulting in data that is more concentrated around the original distribution, while Langevin dynamics yields noisier and more dispersed results.

Following Kim et al. (2022); Ouyang et al. (2023), we conduct experiments on a simulated Bayesian Network dataset and a real Census dataset (Kohavi, 1996) to illustrate the efficiency and effectiveness of the data generated by our proposed model trained on missing data.

**Baselines.** We evaluate the proposed method using both Langevin and Ozaki sampling against several baseline techniques for synthetic data generation on datasets with missing values. Specifically, we implement a vanilla DSM model that (1) removes rows with missing values, and (2) uses mean imputation for missing values in each column. Additionally, we include STaSy (Kim et al., 2022), a state-of-the-art score-based model, which significantly outperforms other approaches for tabular data. As STaSy requires complete datasets for training, we apply mean imputation to handle any missing values in the training data.

**Metrics.** Following Kim et al. (2022); Ouyang et al. (2023), we employ two criteria, $fidelity$ and $utility$, to assess the quality of the generated synthetic tabular data. For evaluating $fidelity$, we utilize the model-agnostic library SDMetrics. The result ranges from 0 to 100%. A higher score indicates better overall quality of the synthetic data. To measure $utility$, we adopt the same pipeline as Kim et al. (2022), training various models—including Decision Tree, AdaBoost, Logistic Regression, MLP Classifier, Random Forest, and XGBoost on the synthetic data and test them with real data. Our primary metric is classification accuracy, and we also report AUROC and Weighted-F1 scores in the Appendix E.4. All experimental results are based on three repetitions.

**Results.** Figure 5 and Table 2 demonstrate the effectiveness of the proposed method on both the simulated Bayesian Network dataset and the Census data, showing superior performance in terms of fidelity and utility compared to other baselines. Specifically, these results confirm that the impute-then-generate approach introduces bias, whereas directly learning from missing data significantly improves the performance of the generative model. Furthermore, the advantages of the proposed model become more pronounced as the missing ratios increase. Additional details and results of the experiments can be found in Appendix E.

Table 2: Fidelity (SDMetric) and Utility (Accuracy) evaluation of MissScore using Langevin and Ozaki samplings, along with other baselines, on the Census dataset with missing ratio 0.3 under MCAR.

|  | Legenvin | Okazi | DSM-delete | DSM-mean | STaSy-mean |
|---|---|---|---|---|---|
| Fiedility | 86% | **88**% | 73% | 77% | 82% |
| Utility | 80% | **81**% | 70% | 75% | 77% |

## 5 CAUSAL DISCOVERY WITH MISSING DATA VIA SECOND-ORDER SCORES

**Background.** Causal discovery aims to identify causal relationship from purely observational data. However, the task is ill-posed without additional assumptions. Assuming an additive noise model (ANM) allows for the identification of causal structures. In this context, consider the ANM defined as $x_i = f_i(x_{\text{PA}_i}) + z_i$, where $f_i$ is a nonlinear function and $z_i$ is a Gaussian noise. Rolland et al. (2022) proposed an order-based algorithm that uses the second order score of an ANM with a probability distribution $p_{\text{data}}(\mathbf{x})$ to identify leaf nodes, and iteratively determine the topological order of

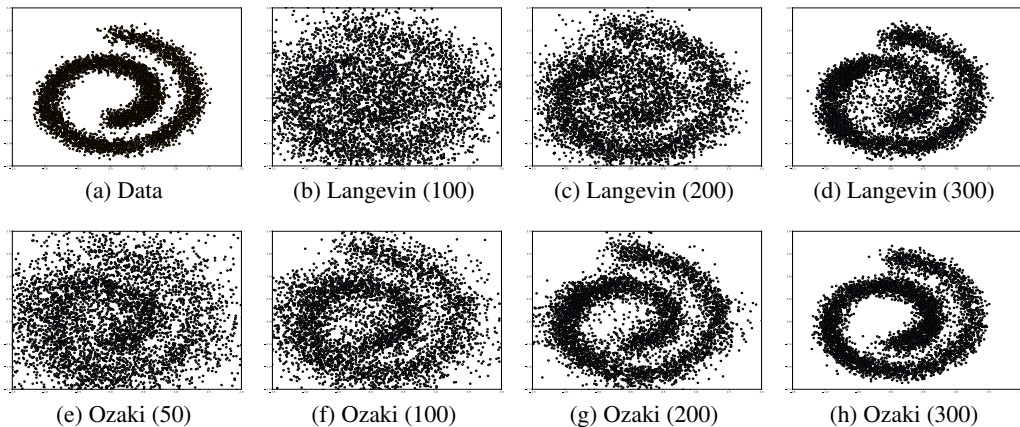

Figure 2: Sampling a Swiss Roll dataset with a step size of 0.005 using Langevin dynamics and Ozaki sampling. Ozaki sampling demonstrates faster convergence and better data quality compared to Langevin dynamics under MCAR with a missing ratio of 0.5. Figure (a) displays the dataset; Figures (b)-(d) show the results from Langevin dynamics; and Figures (e)-(h) present the results from Ozaki sampling. The numbers in parentheses indicate the number of iterations sampling taken.

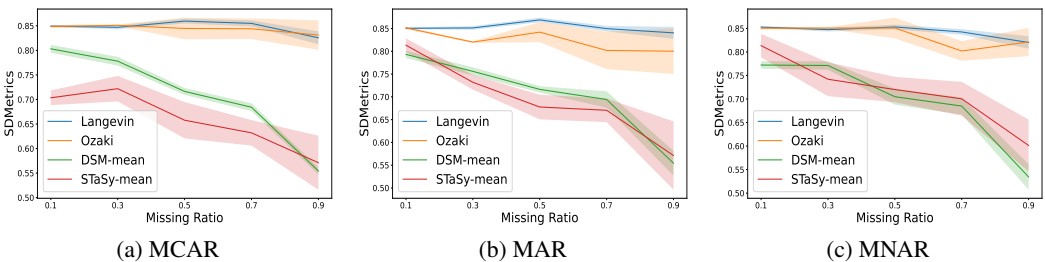

Figure 3: Fidelity evaluation of MissScore using Langevin and Ozaki samplings, along with other baselines, on the Bayesian Network dataset varies with missing ratio $\alpha = \{0.1, 0.3, 0.5, 0.7, 0.9\}$ under different missing mechanisms.

the variables. However, the computation of the Hessian requires complete data, which poses challenges in real-world scenarios such as clinical trials, and biology, where missing data is common.

A straightforward approach to address missing data problem is to first impute the incomplete entries using off-the-shelf imputation methods and then apply existing causal discovery methods. However, this two-step approach can be suboptimal, as the imputation process may introduce bias for modeling the underlying data distribution. Our method mitigates this issue by directly training a second-order model with incomplete data, thereby reducing potential bias. Since the Hessian only provides information about variable order, we adopt a strategy similar to Rolland et al. (2022); Sanchez et al. (2022), first computing the topological order and then using CAM pruning to derive the final directed acyclic graph (DAG) (Bühlmann et al., 2014).

**Baselines.** We utilize the MissForest imputation method to address missing data, followed by the implementation of DiffAN (Sanchez et al., 2022) (termed MissDiffAN) and DAGMA (Bello et al., 2022) (termed MissForest) for structure learning. DiffAN serves as a diffusion-based adaptation of the approach proposed by Rolland et al. (2022), ensuring a fair comparison. Furthermore, we compare MissScore with MissDAG (Gao et al., 2022) and MissOTM (Vo et al., 2024), both of which are prominent methods that have shown superior performance in causal discovery with missing data relative to various other baselines.

**Metrics.** All quantitative results are averaged over 10 random initiazations. For comparing the estimated DAG with the ground-truth one, we report commonly used metrics: Order Divergence and Structural Hamming Distance (SHD). Order Divergence measures the number of errors in the ordering, while SHD indicates the minimum number of edge additions, deletions, and reversals

needed to convert the recovered DAG into the true one. Lower values for both metrics are preferred. Order Divergence is calculated only for MissScore and MissDiffAN, as these are the only order-based methods.

**Simulations.** We simulate synthetic datasets generating a ground-truth DAG from the graph model Erdős–Rényi (ER). Each function $f_i$ is constructed from a multi-layer perceptron (MLP) and a multiple index model (MIM) with random coefficients. We consider a general scenario of non-euqal variances, sampling 1000 observations according to all missing mechansims: MCAR, MAR, MNAR at 10% and 30% missing rates and complete data. In the main text, we report the SHD and runtime in MCAR cases with missing ratio 0.1 varies with number of dimensions, using Gaussian noise. Specifically, the number of edges is set equal to the dimensionality.

**Results.** In Figure 4, our approach demonstrates comparable performance to the state-of-the-art methods MissDAG and MissOTM, although it shows slightly lower performance in the high-dimensional scenario with $d = 50$. This may be attributed to the challenges of training a denoising score matching model with a limited number of samples. However, MissScore significantly reduces computation time compared to MissOTM and MissDAG, offering an efficient alternative while maintaining similar results. With MissForest, our performance is similar, possibly due to the additional constraints enforced during training by DAGMA. However, when comparing our method to those that rely on imputation followed by causal discovery approach, we find that our approach outperforms MissDiffAN. This suggests that imputation can introduce bias in downstream tasks. Additional results for various missing ratios, mechanisms, and order divergences are provided, along with further experimental details, in Appendix F.4.

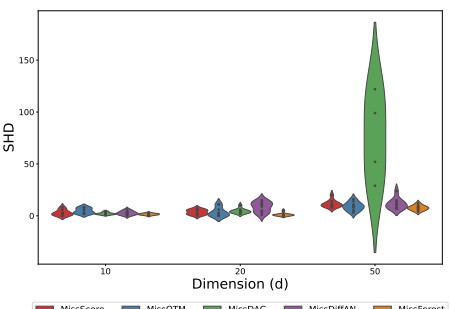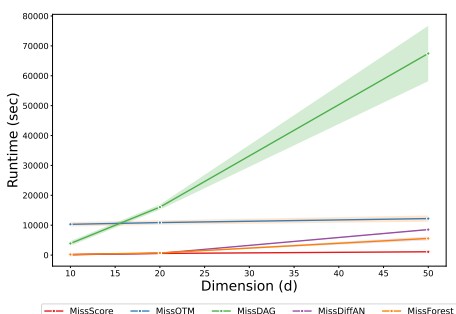

Figure 4: Data is generated under MCAR with a missing ratio of 0.1, varying dimensions $d = \{10, 20, 50\}$ using ER graph model. The sample size is 1000, and $f_i$ corresponds to an MLP. Left: SHD; Right: Runtime. The shaded area indicates 95% confidence.

## 6 CONCLUSION

In this work, we introduce a method to directly estimate higher-order data density scores in the presence of missing data, extending denoising score matching to accommodate scores of any order across different missing data mechanisms. Our approach directly handles missing data without relying on imputation or deletion. Empirical results demonstrate that models trained with this method estimate second-order scores more efficiently and accurately. Moreover, second-order models enhance the sampling quality of Langevin dynamics via Ozaki discretization in missing data scenarios. Our proposed causal discovery method for incomplete data scales effectively with dimensionality and achieves performance comparable to state-of-the-art approaches. However, the effectiveness of this approach diminishes in low-noise environments without variance reduction, particularly for second-order scores, which poses challenges in low-noise or high-missingness scenarios. This limitation also extends to downstream tasks, such as causal discovery and sampling, where performance tends to decline slightly with increasing dimensionality. Future directions include exploring score-based models with constraints for causal discovery in missing data contexts, applying higher-order score estimation to a broader range of applications like image and time-series data, and further investigating the use of denoising score matching for handling MNAR data directly.

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

# Appendix

## Table of Contents

## A    RELATED WORK

**Missing Data**    Addressing missing values in training data has been widely studied, leading to the development of numerous imputation techniques. Traditional approaches often involve either removing rows or columns with missing entries or imputing missing values by substituting them with the mean of observed values for a given feature. More advanced methods apply machine learning and deep generative models for imputation, such as those explored in prior work (Muzellec et al., 2020; Van Buuren & Groothuis-Oudshoorn, 2011; Bertsimas et al., 2018). However, imputation can reduce data diversity and introduce biases in downstream tasks. For instance, Ouyang et al. (2023) highlight that in an "impute-then-generate" pipeline, imputation may impair generation quality by introducing biases. In contrast, "impute-then-predict" pipelines often yield better predictive accuracy for classification and regression tasks, as evidenced by works like (Poulos & Valle, 2018; Jäger et al., 2021; Shadbahr et al., 2023; Paterakis et al., 2024).

Additionally, imputation methods often fail to account for uncertainty in missing data. Multiple imputation (MI) is a valuable alternative that incorporates this uncertainty. While Van Buuren & Groothuis-Oudshoorn (2011) introduce the widely-used MICE algorithm for MI, MI techniques can also be applied with deep learning models such as GAIN and MIDA (Gondara & Wang, 2018; Wang et al., 2021) to produce multiple plausible datasets by re-running models with different random initializations, thereby capturing imputation uncertainty. A comparative study by Wang et al. (2021) indicates that MI with classification trees (like MICE) often outperforms deep learning-based imputation, especially in survey data.

Beyond imputation, recent research has explored models that directly learn from incomplete data or synthesize complete datasets using generative architectures such as GANs and VAEs (Li et al., 2019; Yoon et al., 2018; Park et al., 2018). These approaches typically require auxiliary networks and rely on assumptions about the missing data mechanism. In addition, unique challenges posed by tabular data in these contexts remain underexplored. Recent advancements include applying score-based models with self-paced learning and fine-tuning strategies (Kim et al., 2022), as well as diffusion models, which learn from incomplete data through first-order derivatives (Ouyang et al., 2023).

**Score Matching.**    Score matching estimates the gradient of the log-density (known as the score function) of a data distribution, making them highly effective in modeling complex distributions (Hyvärinen & Dayan, 2005). A prominent technique in this family is Denoising Score Matching, which learns the score of a perturbed version of the target distribution by minimizing a regression loss (Vincent, 2011). This method is also widely used in training Denoising Diffusion Models and has found success in tasks such as image and audio generation (Ho et al., 2020). While first-order score matching is prevalent, higher-order derivatives offer richer information about the data distributions by capturing its local curvature. Although these can be derived using automatic differentiation of a learned score model, such method is computationally expensive and prone to error in high-dimensional settings. To address these, recent work by Meng et al. (2021) extends denoising score matching to estimate higher-order derivatives by leveraging Tweedie's formula, which relates higher-order moments of the distribution to its scores. Lu et al. (2022) further shows that the negative likelihood of the ODE can be bounded by controlling the high-order score matching errors.

## B    ADDITIONAL DISCUSSIONS

**Differences between MissScore and MissDiff**    MissScore differs from MissDiff by focusing on learning high-order score with missing data, specifically leveraging the Hessian to enhance downstream performance. In contrast, MissDiff emphasizes unbiased data synthesis with a first-order diffusion-based approach. Additionally, MissScore introduces an oracle estimator under the MAR mechanism and introduce a novel causal discovery method that operates effectively in the presence of missing data.

**Scalability and numerical stability for MAR**    We have included additional synthetic results for the MAR mechanism in Table 3, following the same setup as described in Section 3.2. The results exhibit a similar pattern, though slightly worse than MCAR, which may be attributed to the inverse probability, potentially increasing variance and leading to less stable estimations. Nevertheless, the estimates remain reasonable and demonstrate empirical performance closely aligned with the

ground truth, along with variance reduction. Additionally, to illustrate the impact of potential model misspecification by the logistic regression model in MAR, we conduct an experiment comparing the ground truth $p$ with the estimated $\hat{p}$ from the logistic regression model on the same synthetic dataset, with varying missing ratios. Comparing the performance in Tables 3 and 4, potential model misspecification in the logistic regression model does impact the estimation of missing probability. However, this effect remains within a reasonable range in the variance-reduced version.

Table 3: Mean squared error (MSE) between the estimated first-order and second-order scores and the ground truth is evaluated across 5,000 test samples. We vary the noise scales $\sigma$ and missing ratios $\alpha$, with each configuration tested using 10 random seeds. **MAR with estimated missing probability $\hat{p}$ by logistic model.**

| Methods | $\alpha = 0.0$ (Complete data) | | $\alpha = 0.1$ | | $\alpha = 0.3$ | | $\alpha = 0.5$ | |
|---|---|---|---|---|---|---|---|---|
| | $\sigma = 0.1$ | $\sigma = 0.01$ | $\sigma = 0.1$ | $\sigma = 0.01$ | $\sigma = 0.1$ | $\sigma = 0.01$ | $\sigma = 0.1$ | $\sigma = 0.01$ |
| $\mathbf{s}_1$ | $0.28 \pm 0.01$ | $0.42 \pm 0.02$ | $0.32 \pm 0.00$ | $0.51 \pm 0.02$ | $0.36 \pm 0.02$ | $0.52 \pm 0.00$ | $0.45 \pm 0.01$ | $0.56 \pm 0.03$ |
| $\mathbf{s}_1$(VR) | $0.07 \pm 0.00$ | $0.07 \pm 0.00$ | $0.11 \pm 0.02$ | $0.13 \pm 0.01$ | $0.15 \pm 0.01$ | $0.16 \pm 0.02$ | $0.22 \pm 0.02$ | $0.24 \pm 0.04$ |
| $\mathbf{s}_2$ | $0.16 \pm 0.02$ | $15.42 \pm 0.47$ | $0.28 \pm 0.02$ | $31.22 \pm 1.08$ | $0.30 \pm 0.04$ | $34.24 \pm 5.14$ | $0.36 \pm 0.04$ | $35.41 \pm 1.03$ |
| $\mathbf{s}_2$(VR) | $0.04 \pm 0.00$ | $0.05 \pm 0.00$ | $0.04 \pm 0.00$ | $0.05 \pm 0.00$ | $0.06 \pm 0.00$ | $0.06 \pm 0.00$ | $0.08 \pm 0.01$ | $0.07 \pm 0.00$ |

Table 4: Mean squared error (MSE) between the estimated first-order and second-order scores and the ground truth is evaluated across 5,000 test samples. We vary the noise scales $\sigma$ and missing ratios $\alpha$, with each configuration tested using 10 random seeds. **MAR with ground truth missing probability $p$.**

| Methods | $\alpha = 0.0$ (Complete data) | | $\alpha = 0.1$ | | $\alpha = 0.3$ | | $\alpha = 0.5$ | |
|---|---|---|---|---|---|---|---|---|
| | $\sigma = 0.1$ | $\sigma = 0.01$ | $\sigma = 0.1$ | $\sigma = 0.01$ | $\sigma = 0.1$ | $\sigma = 0.01$ | $\sigma = 0.1$ | $\sigma = 0.01$ |
| $\mathbf{s}_1$ | $0.28 \pm 0.01$ | $0.42 \pm 0.02$ | $0.24 \pm 0.01$ | $0.42 \pm 0.02$ | $0.27 \pm 0.01$ | $0.41 \pm 0.01$ | $0.33 \pm 0.01$ | $0.43 \pm 0.04$ |
| $\mathbf{s}_1$(VR) | $0.07 \pm 0.00$ | $0.07 \pm 0.00$ | $0.06 \pm 0.00$ | $0.06 \pm 0.00$ | $0.09 \pm 0.00$ | $0.10 \pm 0.00$ | $0.19 \pm 0.00$ | $0.22 \pm 0.01$ |
| $\mathbf{s}_2$ | $0.16 \pm 0.02$ | $15.42 \pm 0.47$ | $0.24 \pm 0.01$ | $16.66 \pm 4.34$ | $0.27 \pm 0.01$ | $41.47 \pm 7.28$ | $0.33 \pm 0.01$ | $39.89 \pm 4.03$ |
| $\mathbf{s}_2$(VR) | $0.04 \pm 0.00$ | $0.05 \pm 0.00$ | $0.02 \pm 0.00$ | $0.03 \pm 0.00$ | $0.03 \pm 0.00$ | $0.03 \pm 0.00$ | $0.05 \pm 0.00$ | $0.05 \pm 0.00$ |

## C  PROOFS

### C.1  PROOF OF THEOREM 3

We know that

$$\mathbb{E}_{\mathbf{x},\mathbf{m}}\mathbb{E}_{\tilde{\mathbf{x}}|\mathbf{x},\mathbf{m}}\left[\left\|\left\{\mathbf{s}(\tilde{\mathbf{x}};\boldsymbol{\theta}) + \frac{1}{\sigma^2}(\tilde{\mathbf{x}} - \mathbf{x})\right\} \odot (\mathbf{1} - \mathbf{m})\right\|_2^2\right]$$

$$=\mathbb{E}_{\tilde{\mathbf{x}},\mathbf{x},\mathbf{m}}\left[\left\|\left\{\mathbf{s}(\tilde{\mathbf{x}};\boldsymbol{\theta}) + \frac{1}{\sigma^2}(\tilde{\mathbf{x}} - \mathbf{x})\right\} \odot (\mathbf{1} - \mathbf{m})\right\|_2^2\right]$$

$$=\mathbb{E}_{\tilde{\mathbf{x}},\mathbf{x}}\mathbb{E}_{m|\tilde{\mathbf{x}},\mathbf{x}}\left[\left\|\left\{\mathbf{s}(\tilde{\mathbf{x}};\boldsymbol{\theta}) + \frac{1}{\sigma^2}(\tilde{\mathbf{x}} - \mathbf{x})\right\} \odot (\mathbf{1} - \mathbf{m})\right\|_2^2\right]$$

$$=\mathbb{E}_{\tilde{\mathbf{x}},\mathbf{x}}\left[\left\|\left\{\mathbf{s}(\tilde{\mathbf{x}};\boldsymbol{\theta}) + \frac{1}{\sigma^2}(\tilde{\mathbf{x}} - \mathbf{x})\right\} \odot \sqrt{\mathbb{E}_{\mathbf{m}|\tilde{\mathbf{x}},\mathbf{x}}(\mathbf{1} - \mathbf{m})}\right\|_2^2\right]$$

$$=\mathbb{E}_{\tilde{\mathbf{x}},\mathbf{x}}\left[\left\|\left\{\mathbf{s}(\tilde{\mathbf{x}};\boldsymbol{\theta}) + \frac{1}{\sigma^2}(\tilde{\mathbf{x}} - \mathbf{x})\right\} \odot \sqrt{\mathbf{w}_1}\right\|_2^2\right].$$

Denote $q_\sigma(\tilde{\mathbf{x}}|\mathbf{x}) = \mathcal{N}(\tilde{\mathbf{x}}; \mathbf{x}, \sigma^2)$, we only need to show there exists some constant $C$ independent of $\boldsymbol{\theta}$ such that

$$\mathbb{E}_{\tilde{\mathbf{x}},\mathbf{x}}\left[\|\{\mathbf{s}(\tilde{\mathbf{x}};\boldsymbol{\theta}) - \nabla_{\tilde{\mathbf{x}}}q_\sigma(\tilde{\mathbf{x}}|\mathbf{x})\} \odot \sqrt{\mathbf{w}_1}\|_2^2\right] = \mathbb{E}_{\tilde{\mathbf{x}},\mathbf{x}}\left[\|\{\mathbf{s}(\tilde{\mathbf{x}};\boldsymbol{\theta}) - \mathbf{s}(\tilde{\mathbf{x}})\} \odot \sqrt{\mathbf{w}_1}\|_2^2\right] + C \quad (12)$$

For simplicity, we consider the case $d = 1$. For the right hand side of equation 12,

$$\text{R.H.S} = w_1 \cdot \left\{\mathbb{E}_{\tilde{x},x}\left[s^2(\tilde{x};\theta)\right] - 2\mathbb{E}_{\tilde{x},x}\left[s(\tilde{x};\theta)s(\tilde{x})\right]\right\} + C.$$

For the left hand side of equation 12,

$$\text{L.H.S} = w_1 \cdot \left\{ \mathbb{E}_{\tilde{x},x} \left[ s^2(\tilde{x};\theta) \right] - 2\mathbb{E}_{\tilde{x},x} \left[ s(\tilde{x};\theta) \nabla_{\tilde{x}} \log q_\sigma(\tilde{x}|x) \right] \right\} + C.$$

Hence, we only need to show

$$\mathbb{E}_{\tilde{x},x} \left[ s(\tilde{x};\theta) s(\tilde{x}) \right] = \mathbb{E}_{\tilde{x},x} \left[ s(\tilde{x};\theta) \nabla_{\tilde{x}} \log q_\sigma(\tilde{x}|x) \right]. \tag{13}$$

We have,

$$\mathbb{E}_{\tilde{x},x} \left[ s(\tilde{x};\theta) s(\tilde{x}) \right] = \int s(\tilde{x};\theta) \nabla_{\tilde{x}} p_{\tilde{x}}(\tilde{x}) d\tilde{x}$$

$$= \int s(\tilde{x};\theta) \nabla_{\tilde{x}} \left( \int p(x) q_\sigma(\tilde{x}|x) dx \right) d\tilde{x}$$

$$= \int \int s(\tilde{x};\theta) p(x) \nabla_{\tilde{x}} \left( q_\sigma(\tilde{x}|x) \right) dx d\tilde{x}$$

$$= \int \int s(\tilde{x};\theta) p(x) q_\sigma(\tilde{x}|x) \nabla_{\tilde{x}} \left( \log q_\sigma(\tilde{x}|x) \right) dx d\tilde{x}$$

$$= \int \int s(\tilde{x};\theta) p_{\tilde{x},x}(\tilde{x},x) \nabla_{\tilde{x}} \left( \log q_\sigma(\tilde{x}|x) \right) dx d\tilde{x} = \mathbb{E}_{\tilde{x},x} \left[ s(\tilde{x};\theta) \nabla_{\tilde{x}} \log q_\sigma(\tilde{x}|x) \right]$$

Hence, we get our desired result.

## C.2 PROOF OF THEOREM 4

We have

$$\mathbb{E}_{\tilde{\mathbf{x}},\mathbf{x},\mathbf{m}} \left[ \left\| \left\{ \mathbf{s}_2(\tilde{\mathbf{x}};\boldsymbol{\theta}) + \mathbf{s}_1(\tilde{\mathbf{x}})\mathbf{s}_1^\top(\tilde{\mathbf{x}}) + \frac{\mathbf{I} - \mathbf{z}\mathbf{z}^\top}{\sigma^2} \right\} \odot \left\{ (\mathbf{1} - \mathbf{m})(\mathbf{1} - \mathbf{m})^\top \right\} \right\|_2^2 \right]$$

$$= \mathbb{E}_{\tilde{\mathbf{x}},\mathbf{x}} \left[ \left\| \left\{ \mathbf{s}_2(\tilde{\mathbf{x}};\boldsymbol{\theta}) + \mathbf{s}_1(\tilde{\mathbf{x}})\mathbf{s}_1^\top(\tilde{\mathbf{x}}) + \frac{\mathbf{I} - \mathbf{z}\mathbf{z}^\top}{\sigma^2} \right\} \odot \sqrt{\mathbb{E}_{\mathbf{m}|\tilde{\mathbf{x}},\mathbf{x}}(\mathbf{1} - \mathbf{m})(\mathbf{1} - \mathbf{m})^\top} \right\|_2^2 \right]$$

$$= \mathbb{E}_{\tilde{\mathbf{x}},\mathbf{x}} \left[ \left\| \left\{ \mathbf{s}_2(\tilde{\mathbf{x}};\boldsymbol{\theta}) + \mathbf{s}_1(\tilde{\mathbf{x}})\mathbf{s}_1^\top(\tilde{\mathbf{x}}) + \frac{\mathbf{I} - \mathbf{z}\mathbf{z}^\top}{\sigma^2} \right\} \odot \sqrt{\mathbf{w}_2} \right\|_2^2 \right]$$

It is sufficient to show

$$\mathbb{E}_{\tilde{\mathbf{x}},\mathbf{x}} \left[ \left\| \left\{ \mathbf{s}_2(\tilde{\mathbf{x}};\boldsymbol{\theta}) + \mathbf{s}_1(\tilde{\mathbf{x}})\mathbf{s}_1^\top(\tilde{\mathbf{x}}) + \frac{\mathbf{I} - \mathbf{z}\mathbf{z}^\top}{\sigma^2} \right\} \odot \sqrt{\mathbf{w}_2} \right\|_2^2 \right]$$

$$= \mathbb{E}_{\mathbf{x}} \mathbb{E}_{\tilde{\mathbf{x}}|\mathbf{x}} \left[ \| \{ \mathbf{s}_2(\tilde{\mathbf{x}};\boldsymbol{\theta}) - \mathbf{s}_2(\tilde{\mathbf{x}}) \} \odot \sqrt{\mathbf{w}_2} \|_2^2 \right] + C. \tag{14}$$

with some constant $C$ independent of $\boldsymbol{\theta}$.

For the right hand side of equation 14,

$$\mathbb{E}_{\mathbf{x}} \mathbb{E}_{\tilde{\mathbf{x}}|\mathbf{x}} \left[ \| \{ \mathbf{s}_2(\tilde{\mathbf{x}};\boldsymbol{\theta}) - \mathbf{s}_2(\tilde{\mathbf{x}}) \} \odot \sqrt{\mathbf{w}_2} \|_2^2 \right]$$

$$= \sum_{i=1}^d \sum_{j=1}^d w_{2,ij} \cdot \left\{ \mathbb{E}_{\tilde{x},x} \left[ s_{2,ij}^2(\tilde{x};\theta) \right] - 2\mathbb{E}_{\tilde{x},x} \left[ s_{2,ij}(\tilde{x};\theta) s_{2,ij}(\tilde{x}) \right] + \mathbb{E}_{\tilde{x},x} \left[ s_{2,ij}^2(\tilde{x}) \right] \right\}$$

$$= \sum_{i=1}^d \sum_{j=1}^d w_{2,ij} \cdot \left\{ \mathbb{E}_{\tilde{x},x} \left[ s_{2,ij}^2(\tilde{x};\theta) \right] - 2\mathbb{E}_{\tilde{x},x} \left[ s_{2,ij}(\tilde{x};\theta) s_{2,ij}(\tilde{x}) \right] + \mathbb{E}_{\tilde{x},x} \left[ s_{2,ij}^2(\tilde{x}) \right] \right\} + C_1$$

where $C_1$ is some constant independent of $\boldsymbol{\theta}$.

For the left hand side of equation 14, note that $\frac{\mathbf{I}-\mathbf{z}\mathbf{z}^\top}{\sigma^2} = -\left\{\nabla_{\tilde{x}}^2 q_\sigma(\tilde{x}|x)\right\} - \nabla_{\tilde{x}} q_\sigma(\tilde{x}|x) \cdot \left\{\nabla_{\tilde{x}} q_\sigma(\tilde{x}|x)\right\}^\top$, then we know that

$$\mathbb{E}_{\tilde{\mathbf{x}},\mathbf{x}}\left[\left\|\left\{\mathbf{s}_2(\tilde{\mathbf{x}};\boldsymbol{\theta}) + \mathbf{s}_1(\tilde{\mathbf{x}})\mathbf{s}_1^\top(\tilde{\mathbf{x}}) + \frac{\mathbf{I}-\mathbf{z}\mathbf{z}^\top}{\sigma^2}\right\} \odot \sqrt{\mathbf{w}_2}\right\|_2^2\right]$$

$$= \sum_{i=1}^d \sum_{j=1}^d w_{2,ij}\left(\mathbb{E}_{\tilde{x},x}\left[s_{2,ij}^2(\tilde{x};\theta)\right] + 2\mathbb{E}\left[s_{2,ij}^2(\tilde{x};\theta)s_{1,i}(\tilde{x})s_{1,j}(\tilde{x})\right]\right.$$

$$- 2\mathbb{E}\left[s_{2,ij}(\tilde{x};\theta)\nabla_{\tilde{x}_i}\nabla_{\tilde{x}_j}\log q_\sigma(\tilde{x}|x)\right]$$

$$\left. - 2\mathbb{E}\left[s_{2,ij}(\tilde{x};\theta)\nabla_{\tilde{x}_i}q_\sigma(\tilde{x}|x)\nabla_{\tilde{x}_j}q_\sigma(\tilde{x}|x)\right]\right) + C_2.$$

Comparing left and right hand side of equation 14, it is sufficient to show

$$\mathbb{E}_{\tilde{x},x}\left[s_{2,ij}(\tilde{x};\theta)s_{2,ij}(\tilde{x})\right] = \mathbb{E}\left[s_{2,ij}(\tilde{x};\theta)\nabla_{\tilde{x}_i}\nabla_{\tilde{x}_j}\log q_\sigma(\tilde{x}|x)\right]$$
$$+ \mathbb{E}\left[s_2(\tilde{x};\theta)\nabla_{\tilde{x}_i}q_\sigma(\tilde{x}|x)\nabla_{\tilde{x}_j}q_\sigma(\tilde{x}|x)\right] - \mathbb{E}\left[s_2(\tilde{x};\theta)s_1^2(\tilde{x})\right]. \quad (15)$$

We have

$$\mathbb{E}_{\tilde{x},x}\left[s_{2,ij}(\tilde{x};\theta)s_{2,ij}(\tilde{x})\right] = \int s_{2,ij}(\tilde{x};\theta)s_{2,ij}(\tilde{x})p_{\tilde{x}}(\tilde{x})d\tilde{x}$$

$$= \int s_{2,ij}(\tilde{x};\theta)\left(\nabla_{\tilde{x}_i}\left\{\frac{\nabla_{\tilde{x}_j}p(\tilde{x})}{p(\tilde{x})}\right\}\right)p_{\tilde{x}}(\tilde{x})d\tilde{x}dx$$

$$= \int s_{2,ij}(\tilde{x};\theta)\left\{\frac{\nabla_{\tilde{x}_i}\nabla_{\tilde{x}_j}p(\tilde{x})}{p(\tilde{x})} - \frac{\nabla_{\tilde{x}_i}p(\tilde{x})\cdot\nabla_{\tilde{x}_j}p(\tilde{x})}{p^2(\tilde{x})}\right\}p_{\tilde{x}}(\tilde{x})d\tilde{x}$$

$$= \int s_2(\tilde{x};\theta)\nabla_{\tilde{x}_i}\nabla_{\tilde{x}_j}p(\tilde{x})d\tilde{x} \quad (16)$$

$$- \int s_2(\tilde{x};\theta)\frac{\nabla_{\tilde{x}_i}p(\tilde{x})\cdot\nabla_{\tilde{x}_j}p(\tilde{x})}{p(\tilde{x})}d\tilde{x}. \quad (17)$$

**For equation 16:**

$$\int s_2(\tilde{x};\theta)\nabla_{\tilde{x}_i}\nabla_{\tilde{x}_j}p_{\tilde{x}}(\tilde{x})d\tilde{x} = \int s_2(\tilde{x};\theta)\nabla_{\tilde{x}_i}\nabla_{\tilde{x}_j}\left\{\int q_\sigma(\tilde{x}|x)p_x(x)dx\right\}d\tilde{x}$$

$$= \iint s_2(\tilde{x};\theta)\left\{\nabla_{\tilde{x}_i}\nabla_{\tilde{x}_j}q_\sigma(\tilde{x}|x)\right\}\cdot p_x(x)d\tilde{x}dx$$

$$= \iint s_2(\tilde{x};\theta)]\left\{\nabla_{\tilde{x}_i}\nabla_{\tilde{x}_j}\log q_\sigma(\tilde{x}|x)\right\}\cdot q_\sigma(\tilde{x}|x)p_x(x)d\tilde{x}dx$$

$$+ \iint s_2(\tilde{x};\theta)\frac{\nabla_{\tilde{x}_i}q_\sigma(\tilde{x}|x)\nabla_{\tilde{x}_j}q_\sigma(\tilde{x}|x)}{q_\sigma(\tilde{x}|x)}p_x(x)d\tilde{x}dx$$

$$= \iint s_2(\tilde{x};\theta)\left\{\nabla_{\tilde{x}_i}\nabla_{\tilde{x}_j}\log q_\sigma(\tilde{x}|x)\right\}p_{x,\tilde{x}}(x,\tilde{x})d\tilde{x}dx$$

$$+ \iint s_2(\tilde{x};\theta)\left\{\frac{\nabla_{\tilde{x}_i}q_\sigma(\tilde{x}|x)}{q_\sigma(\tilde{x}|x)}\right\}\left\{\frac{\nabla_{\tilde{x}_j}q_\sigma(\tilde{x}|x)}{q_\sigma(\tilde{x}|x)}\right\}p_{x,\tilde{x}}(x,\tilde{x})d\tilde{x}dx$$

$$= \mathbb{E}\left[s_2(\tilde{x};\theta)\nabla_{\tilde{x}_i}\nabla_{\tilde{x}_j}\log q_\sigma(\tilde{x}|x)\right]$$
$$+ \mathbb{E}\left[s_2(\tilde{x};\theta)\nabla_{\tilde{x}_i}q_\sigma(\tilde{x}|x)\nabla_{\tilde{x}_j}q_\sigma(\tilde{x}|x)\right].$$

**For equation 17:**

$$\int s_2(\tilde{x};\theta)\frac{\nabla_{\tilde{x}_i}p(\tilde{x})\cdot\nabla_{\tilde{x}_j}p(\tilde{x})}{p(\tilde{x})}d\tilde{x} = \int s_2(\tilde{x};\theta))\frac{\nabla_{\tilde{x}_i}p(\tilde{x})}{p(\tilde{x})}\frac{\nabla_{\tilde{x}_j}p(\tilde{x})}{p(\tilde{x})}p(\tilde{x})d\tilde{x}$$

$$= \mathbb{E}\left[s_2(\tilde{x};\theta)s_1^2(\tilde{x})\right].$$

Combine all the results, we get the desired result.

## C.3 Proof of Theorem 7

We first consider the MCAR case. Recall that $\mathbf{w} = \mathbf{1} - \mathbf{m}$

$$\mathbb{E}_{\tilde{\mathbf{x}}, \mathbf{x}, \mathbf{m}} \left[ \| \left\{ \otimes^k \mathbf{x} - f_k(\tilde{\mathbf{x}}, \mathbf{s}_1(\tilde{\mathbf{x}}), \ldots, \mathbf{s}_{k-1}(\tilde{\mathbf{x}}), \mathbf{s}_k(\tilde{\mathbf{x}}; \boldsymbol{\theta})) \right\} \odot \otimes^k \mathbf{w} \|^2 \right]$$

$$= \mathbb{E}_{\tilde{\mathbf{x}}, \mathbf{x}, \mathbf{m}} \left[ \| \left\{ \otimes^k \mathbf{x} - f_k(\tilde{\mathbf{x}}, \mathbf{s}_1(\tilde{\mathbf{x}}), \ldots, \mathbf{s}_{k-1}(\tilde{\mathbf{x}}), \mathbf{s}_k(\tilde{\mathbf{x}}; \boldsymbol{\theta})) \right\} \odot \otimes^k (\mathbf{1} - \mathbf{m}) \|^2 \right]$$

$$= \mathbb{E}_{\tilde{\mathbf{x}}, \mathbf{x}} \mathbb{E}_{\mathbf{m} | \tilde{\mathbf{x}}, \mathbf{x}} \left[ \| \left\{ \otimes^k \mathbf{x} - f_k(\tilde{\mathbf{x}}, \mathbf{s}_1(\tilde{\mathbf{x}}), \ldots, \mathbf{s}_{k-1}(\tilde{\mathbf{x}}), \mathbf{s}_k(\tilde{\mathbf{x}}; \boldsymbol{\theta})) \right\} \odot \otimes^k (\mathbf{1} - \mathbf{m}) \|^2 \right]$$

$$= \mathbb{E}_{\tilde{\mathbf{x}}, \mathbf{x}} \left[ \| \left\{ \otimes^k \mathbf{x} - f_k(\tilde{\mathbf{x}}, \mathbf{s}_1(\tilde{\mathbf{x}}), \ldots, \mathbf{s}_{k-1}(\tilde{\mathbf{x}}), \mathbf{s}_k(\tilde{\mathbf{x}}; \boldsymbol{\theta})) \right\} \odot \left\{ \mathbb{E}_{\mathbf{m}} \otimes^k (\mathbf{1} - \mathbf{m}) \right\} \|^2 \right]$$

Noting that $\mathbb{E}_{\mathbf{m}} \otimes^k (\mathbf{1} - \mathbf{m})$ is a constant. We can show that the solution for the weighted least square equation for any constant matrix $\mathbf{c}$ is

$$\arg\min_h \mathbb{E} \left[ \| \left\{ \otimes^k \mathbf{x} - h(\tilde{\mathbf{x}}) \right\} \odot \mathbf{c} \|^2 \right] = \mathbb{E} \left\{ \otimes^k \mathbf{x} \mid \tilde{\mathbf{x}} \right\}.$$

Such equation holds since

$$\mathbb{E} \left[ \| \left\{ \otimes^k \mathbf{x} - h(\tilde{\mathbf{x}}) \right\} \odot \mathbf{c} \|^2 \right] = \mathbb{E} \left[ \| \left\{ \otimes^k \mathbf{x} - \mathbb{E} \left[ \otimes^k \mathbf{x} \mid \tilde{\mathbf{x}} \right] \right\} \odot \mathbf{c} \|^2 \right]$$

$$+ \mathbb{E} \left[ \| \left\{ \mathbb{E} \left[ \otimes^k \mathbf{x} \mid \tilde{\mathbf{x}} \right] - h(\tilde{\mathbf{x}}) \right\} \odot \mathbf{c} \|^2 \right]$$

$$\geq \mathbb{E} \left[ \| \left\{ \otimes^k \mathbf{x} - \mathbb{E} \left[ \otimes^k \mathbf{x} \mid \tilde{\mathbf{x}} \right] \right\} \odot \mathbf{c} \|^2 \right]$$

By Theorem 2, we know that $\mathbb{E} \left\{ \otimes^k \mathbf{x} \mid \tilde{\mathbf{x}} \right\} = f_k(\tilde{\mathbf{x}}, \mathbf{s}_1(\tilde{\mathbf{x}}), \ldots, \mathbf{s}_{k-1}(\tilde{\mathbf{x}}), \mathbf{s}_k(\tilde{\mathbf{x}}))$. Hence, the desired result follows.

For the MAR case, recall that $\mathbf{w} = \frac{\mathbf{1} - \mathbf{m}}{\mathbb{P}[\mathbf{m}^k = 0 | \mathbf{x} = \mathbf{x}]}$

$$\mathbb{E}_{\tilde{\mathbf{x}}, \mathbf{x}, \mathbf{m}} \left[ \| \left\{ \otimes^k \mathbf{x} - f_k(\tilde{\mathbf{x}}, \mathbf{s}_1(\tilde{\mathbf{x}}), \ldots, \mathbf{s}_{k-1}(\tilde{\mathbf{x}}), \mathbf{s}_k(\tilde{\mathbf{x}}; \boldsymbol{\theta})) \right\} \odot \otimes^k \mathbf{w} \|^2 \right]$$

$$= \mathbb{E}_{\tilde{\mathbf{x}}, \mathbf{x}, \mathbf{m}} \left[ \left\| \left\{ \otimes^k \mathbf{x} - f_k(\tilde{\mathbf{x}}, \mathbf{s}_1(\tilde{\mathbf{x}}), \ldots, \mathbf{s}_{k-1}(\tilde{\mathbf{x}}), \mathbf{s}_k(\tilde{\mathbf{x}}; \boldsymbol{\theta})) \right\} \odot \otimes^k \frac{\mathbf{1} - \mathbf{m}}{\mathbb{P}[\mathbf{m}^k = 0 | \mathbf{x} = \mathbf{x}]} \right\|^2 \right]$$

$$= \mathbb{E}_{\tilde{\mathbf{x}}, \mathbf{x}} \mathbb{E}_{\mathbf{m} | \mathbf{x}} \left[ \left\| \left\{ \otimes^k \mathbf{x} - f_k(\tilde{\mathbf{x}}, \mathbf{s}_1(\tilde{\mathbf{x}}), \ldots, \mathbf{s}_{k-1}(\tilde{\mathbf{x}}), \mathbf{s}_k(\tilde{\mathbf{x}}; \boldsymbol{\theta})) \right\} \odot \otimes^k \frac{\mathbf{1} - \mathbf{m}}{\mathbb{P}[\mathbf{m}^k = 0 | \mathbf{x} = \mathbf{x}]} \right\|^2 \right]$$

$$= \mathbb{E}_{\tilde{\mathbf{x}}, \mathbf{x}} \left[ \| \left\{ \otimes^k \mathbf{x} - f_k(\tilde{\mathbf{x}}, \mathbf{s}_1(\tilde{\mathbf{x}}), \ldots, \mathbf{s}_{k-1}(\tilde{\mathbf{x}}), \mathbf{s}_k(\tilde{\mathbf{x}}; \boldsymbol{\theta})) \right\} \|^2 \right].$$

Thus, by Theorem 2, the solution to the criterion function is

$$\arg\min_\theta \mathbb{E}_{\tilde{\mathbf{x}}, \mathbf{x}, \mathbf{m}} \left[ \| \left\{ \otimes^k \mathbf{x} - f_k(\tilde{\mathbf{x}}, \mathbf{s}_1(\tilde{\mathbf{x}}), \ldots, \mathbf{s}_{k-1}(\tilde{\mathbf{x}}), \mathbf{s}_k(\tilde{\mathbf{x}}; \boldsymbol{\theta})) \right\} \odot \otimes^k \mathbf{w} \|^2 \right] = \theta^*.$$

## C.4 Proof of Theorem 5

By similar argument in the proof of Theorem 3,

$$\mathbb{E}_{\mathbf{x}, \mathbf{m}} \mathbb{E}_{\tilde{\mathbf{x}} | \mathbf{x}, \mathbf{m}} \left[ \left\| \left\{ \mathbf{s}(\tilde{\mathbf{x}}; \boldsymbol{\theta}) + \frac{1}{\sigma^2} (\tilde{\mathbf{x}} - \mathbf{x}) \right\} \odot \frac{\mathbf{1} - \mathbf{m}}{\sqrt{\mathbb{P}[\mathbf{m} = 0 | \mathbf{x} = \mathbf{x}]}} \right\|_2^2 \right]$$

$$= \mathbb{E}_{\tilde{\mathbf{x}}, \mathbf{x}, \mathbf{m}} \left[ \left\| \left\{ \mathbf{s}(\tilde{\mathbf{x}}; \boldsymbol{\theta}) + \frac{1}{\sigma^2} (\tilde{\mathbf{x}} - \mathbf{x}) \right\} \odot \frac{\mathbf{1} - \mathbf{m}}{\sqrt{\mathbb{P}[\mathbf{m} = 0 | \mathbf{x} = \mathbf{x}]}} \right\|_2^2 \right]$$

$$= \mathbb{E}_{\tilde{\mathbf{x}}, \mathbf{x}} \mathbb{E}_{m | \tilde{\mathbf{x}}, \mathbf{x}} \left[ \left\| \left\{ \mathbf{s}(\tilde{\mathbf{x}}; \boldsymbol{\theta}) + \frac{1}{\sigma^2} (\tilde{\mathbf{x}} - \mathbf{x}) \right\} \odot \frac{\mathbf{1} - \mathbf{m}}{\sqrt{\mathbb{P}[\mathbf{m} = 0 | \mathbf{x} = \mathbf{x}]}} \right\|_2^2 \right]$$

$$= \mathbb{E}_{\tilde{\mathbf{x}}, \mathbf{x}} \left[ \left\| \left\{ \mathbf{s}(\tilde{\mathbf{x}}; \boldsymbol{\theta}) + \frac{1}{\sigma^2} (\tilde{\mathbf{x}} - \mathbf{x}) \right\} \odot \frac{\sqrt{\mathbb{E}_{m | \tilde{\mathbf{x}}, \mathbf{x}} (\mathbf{1} - \mathbf{m})}}{\sqrt{\mathbb{P}[\mathbf{m} = 0 | \mathbf{x} = \mathbf{x}]}} \right\|_2^2 \right]$$

$$= \mathbb{E}_{\tilde{\mathbf{x}}, \mathbf{x}} \left[ \left\| \left\{ \mathbf{s}(\tilde{\mathbf{x}}; \boldsymbol{\theta}) + \frac{1}{\sigma^2} (\tilde{\mathbf{x}} - \mathbf{x}) \right\} \right\|_2^2 \right].$$

where the last equality comes from $\mathbb{E}_{\mathbf{m}|\tilde{\mathbf{x}},\mathbf{x}}(\mathbf{1} - \mathbf{m}) = \mathbb{E}_{\mathbf{m}|\mathbf{x}}(\mathbf{1} - \mathbf{m}) = \mathbb{P}[\mathbf{m} = 0|\mathbf{x} = \mathbf{x}]$. Then following the steps in section C.1, taking $\mathbf{w}_1$ in C.1 as a vector of 1, we get our desired result.

## C.5 PROOF OF THEOREM 6

By similar argument in the proof of Theorem 4,

$$
\mathbb{E}_{\tilde{\mathbf{x}},\mathbf{x},\mathbf{m}}\left[\left\|\left\{\mathbf{s}_2(\tilde{\mathbf{x}};\boldsymbol{\theta}) + \mathbf{s}_1(\tilde{\mathbf{x}})\mathbf{s}_1^\top(\tilde{\mathbf{x}}) + \frac{\mathbf{I} - \mathbf{z}\mathbf{z}^\top}{\sigma^2}\right\} \odot \frac{\{(\mathbf{1}-\mathbf{m})(\mathbf{1}-\mathbf{m})^\top\}}{\sqrt{\mathbb{P}[\mathbf{m}\mathbf{m}^\top = 0|\mathbf{x} = \mathbf{x}]}}\right\|_2^2\right]
$$

$$
=\mathbb{E}_{\tilde{\mathbf{x}},\mathbf{x}}\left[\left\|\left\{\mathbf{s}_2(\tilde{\mathbf{x}};\boldsymbol{\theta}) + \mathbf{s}_1(\tilde{\mathbf{x}})\mathbf{s}_1^\top(\tilde{\mathbf{x}}) + \frac{\mathbf{I} - \mathbf{z}\mathbf{z}^\top}{\sigma^2}\right\} \odot \frac{\sqrt{\mathbb{E}_{\mathbf{m}|\tilde{\mathbf{x}},\mathbf{x}}(\mathbf{1}-\mathbf{m})(\mathbf{1}-\mathbf{m})^\top}}{\sqrt{\mathbb{P}[\mathbf{m}\mathbf{m}^\top = 0|\mathbf{x} = \mathbf{x}]}}\right\|_2^2\right]
$$

$$
=\mathbb{E}_{\tilde{\mathbf{x}},\mathbf{x}}\left[\left\|\left\{\mathbf{s}_2(\tilde{\mathbf{x}};\boldsymbol{\theta}) + \mathbf{s}_1(\tilde{\mathbf{x}})\mathbf{s}_1^\top(\tilde{\mathbf{x}}) + \frac{\mathbf{I} - \mathbf{z}\mathbf{z}^\top}{\sigma^2}\right\}\right\|_2^2\right].
$$

where the last equality comes from $\mathbb{E}_{\mathbf{m}|\tilde{\mathbf{x}},\mathbf{x}}(\mathbf{1} - \mathbf{m})(\mathbf{1} - \mathbf{m})^\top = \mathbb{E}_{\mathbf{m}|\mathbf{x}}(\mathbf{1} - \mathbf{m})(\mathbf{1} - \mathbf{m})^\top = \mathbb{P}[\mathbf{m}\mathbf{m}^\top = 0|\mathbf{x} = \mathbf{x}]$. Then following the steps in section C.2, taking $\mathbf{w}_2$ in C.2 as $\mathbf{1}$, we get our desired result.

## C.6 PROOF OF EQUATION 8

Denote the oracle criterion function as

$$
\widetilde{\mathcal{L}}_{\text{DSM}}(\boldsymbol{\theta}) := \mathbb{E}_{\mathbf{x},\mathbf{m}}\mathbb{E}_{\tilde{\mathbf{x}}|\mathbf{x},\mathbf{m}}\left[\left\|\left\{\mathbf{s}_1(\tilde{\mathbf{x}};\boldsymbol{\theta}) + \frac{1}{\sigma^2}(\tilde{\mathbf{x}} - \mathbf{x})\right\} \odot \mathbf{g}_1(\mathbf{x},\mathbf{m})\right\|_2^2\right].
$$

where $\mathbf{g}_1(\mathbf{x},\mathbf{m}) = 1 - \mathbf{m}$ under MAR and $\mathbf{g}_1(\mathbf{x},\mathbf{m}) = \frac{1-\mathbf{m}}{\sqrt{\mathbb{P}[\mathbf{m}=0|\mathbf{x}=\mathbf{x}]}}$.

If we want to match the score of true data distribution $p(\mathbf{x})$, $\sigma$ should be approximately zero for both DSM and $D_2$SM so that $q_\sigma(\tilde{\mathbf{x}})$ is close to $p(\mathbf{x})$. According to Taylor expansion we have,

$$
\widetilde{\mathcal{L}}_{\text{DSM}}(\boldsymbol{\theta}) = \mathbb{E}_{\tilde{\mathbf{x}},\mathbf{x},\mathbf{m}}\left[\left\|\left\{\mathbf{s}_1(\tilde{\mathbf{x}};\boldsymbol{\theta}) + \frac{1}{\sigma^2}(\tilde{\mathbf{x}} - \mathbf{x})\right\} \odot \mathbf{g}_1(\mathbf{x},\mathbf{m})\right\|_2^2\right]
$$

$$
= \mathbb{E}_{\mathbf{x},\mathbf{m}}\mathbb{E}_{\mathbf{z}\sim\mathcal{N}(\mathbf{0},\mathbf{I})}\left[\left\|\left\{\mathbf{s}_1(\mathbf{x} + \sigma\mathbf{z};\boldsymbol{\theta}) + \frac{\mathbf{z}}{\sigma}\right\} \odot \mathbf{g}_1(\mathbf{x},\mathbf{m})\right\|_2^2\right]
$$

$$
= \mathbb{E}_{\mathbf{x},\mathbf{m}}\mathbb{E}_{\mathbf{z}\sim\mathcal{N}(\mathbf{0},\mathbf{I})}\left[\left\|\left\{\mathbf{s}_1(\mathbf{x};\boldsymbol{\theta}) + \sigma\nabla_\mathbf{x}\mathbf{s}_1(\mathbf{x};\boldsymbol{\theta})\mathbf{z} + \frac{\mathbf{z}}{\sigma}\right\} \odot \mathbf{g}_1(\mathbf{x},\mathbf{m})\right\|_2^2\right] + \mathcal{O}(1)
$$

$$
= \mathbb{E}_{\mathbf{x},\mathbf{m}}\mathbb{E}_{\mathbf{z}\sim\mathcal{N}(\mathbf{0},\mathbf{I})}\left[\left\|\left\{\mathbf{s}_1(\mathbf{x};\boldsymbol{\theta}) + \frac{\mathbf{z}}{\sigma}\right\} \odot \mathbf{g}_1(\mathbf{x},\mathbf{m})\right\|_2^2\right] + \mathcal{O}(1)
$$

$$
= \mathbb{E}_{\mathbf{x},\mathbf{m}}\mathbb{E}_{\mathbf{z}\sim\mathcal{N}(\mathbf{0},\mathbf{I})}\left[\left\{\|\mathbf{s}_1(\mathbf{x};\boldsymbol{\theta}) \odot \mathbf{g}_1(\mathbf{x})\|_2^2 + \frac{\|\mathbf{z} \odot \mathbf{g}_1(\mathbf{x},\mathbf{m})\|^2}{\sigma^2} + \left(\frac{2}{\sigma}\mathbf{s}_1(\mathbf{x};\boldsymbol{\theta})^\top\mathbf{z}\right) \odot \mathbf{g}_1(\mathbf{x},\mathbf{m})\right\}\right] + \mathcal{O}(1)
$$

where $\mathbf{z} = \frac{\tilde{\mathbf{x}}-\mathbf{x}}{\sigma}$, where $\mathcal{O}(1)$ is bounded as $\sigma$ approaches zero. However, when evaluating the expectation above from samples, the variances of $\frac{\|\mathbf{z}\odot\mathbf{g}(\mathbf{x},\mathbf{m})\|^2}{\sigma^2}$ and $\frac{\mathbf{s}(\mathbf{x};\boldsymbol{\theta})^\top\mathbf{z}}{\sigma}$ both increase without bound as $\sigma$ nears zero, due to the terms involving $\sigma$ and $\sigma^2$ in the denominator. This leads to a significant increase in the variance of the DSM loss, complicating the optimization process. As a consequence, DSM may become unstable and fail to converge when $\sigma$ is small, highlighting the need for methods to reduce variance.

We have,

$$
\mathbb{E}_{\mathbf{z}\sim\mathcal{N}(\mathbf{0},\mathbf{I})}\left[\frac{\|\mathbf{z} \odot \mathbf{g}_1(\mathbf{x},\mathbf{m})\|^2}{\sigma^2} + \frac{2}{\sigma}\mathbf{s}_1(\mathbf{x};\boldsymbol{\theta})^\top\mathbf{z}\right] = \frac{\|\mathbf{g}_1(\mathbf{x},\mathbf{m})\|^2}{\sigma^2},
$$

where $d$ is the dimension of the data distribution $p(\mathbf{x})$. Therefore, we can construct a variable that is, for sufficiently small $\sigma$, positively correlated with $\mathcal{L}_{\text{DSM}}$ while having an expected value of zero:

$$c_{\boldsymbol{\theta}}(\mathbf{x}; \mathbf{z}) = \left(\frac{2}{\sigma}\mathbf{s}_1(\mathbf{x}; \boldsymbol{\theta})^\top \mathbf{z}\right) \odot \mathbf{g}_1(\mathbf{x}, \mathbf{m}) + \frac{\|\mathbf{z} \odot \mathbf{g}_1(\mathbf{x}, \mathbf{m})\|^2}{\sigma^2} - \frac{\|\mathbf{g}_1(\mathbf{x}, \mathbf{m})\|^2}{\sigma^2}.$$

Subtracting it from $\mathcal{L}_{\text{DSM}}$ will yield an estimator with reduced variance for DSM training with missing data:

$$\mathcal{L}_{\text{DSM-VR}}(\boldsymbol{\theta}) = \mathcal{L}_{\text{DSM}}(\boldsymbol{\theta}) - \mathbb{E}_{\mathbf{x}, \mathbf{m}} \mathbb{E}_{\mathbf{z} \sim \mathcal{N}(\mathbf{0}, \mathbf{I})} \left[ \left(\frac{2}{\sigma}\mathbf{s}(\mathbf{x}; \boldsymbol{\theta})^\top \mathbf{z}\right) \odot \mathbf{g}_1(\mathbf{x}, \mathbf{m}) + \frac{\|\mathbf{z} \odot \mathbf{g}_1(\mathbf{x}, \mathbf{m})\|^2}{\sigma^2} \right].$$

Here we omit the part $\frac{\|\mathbf{g}_1(\mathbf{x}, \mathbf{m})\|^2}{\sigma^2}$ since it is independent of $\boldsymbol{\theta}$.

## C.7 Proof of Equation 9

Similar to proof C.6, consider the oracle criterion function

$$\widetilde{L}_{\text{D}_2\text{SM}}(\boldsymbol{\theta}) = \mathbb{E}_{\tilde{\mathbf{x}}, \mathbf{x}, \mathbf{m}} \left[ \left\| \left\{ \mathbf{s}_2(\tilde{\mathbf{x}}; \boldsymbol{\theta}) + \mathbf{s}_1(\tilde{\mathbf{x}})\mathbf{s}_1^\top(\tilde{\mathbf{x}}) + \frac{\mathbf{I} - \mathbf{z}\mathbf{z}^\top}{\sigma^2} \right\} \odot \mathbf{g}_2(\mathbf{x}, \mathbf{m}) \right\|_2^2 \right],$$

where $\mathbf{g}_2(\mathbf{x}_{\text{obs}}, \mathbf{m}) = (\mathbf{1} - \mathbf{m})(\mathbf{1} - \mathbf{m})^\top$ under MCAR, and $\mathbf{g}_2(\mathbf{x}_{\text{obs}}, \mathbf{m}) = \frac{(\mathbf{1} - \mathbf{m})(\mathbf{1} - \mathbf{m})^\top}{\sqrt{\mathbb{E}[\mathbf{m}\mathbf{m}^\top | \mathbf{x} = \mathbf{x}_{\text{obs}}]}}$ under MAR.

Denote $\boldsymbol{\psi}(\tilde{\mathbf{x}}; \boldsymbol{\theta}) = \mathbf{s}_2(\tilde{\mathbf{x}}_{\text{obs}}; \boldsymbol{\theta}) + \mathbf{s}_1(\tilde{\mathbf{x}}_{\text{obs}})\mathbf{s}_1^\top(\tilde{\mathbf{x}}_{\text{obs}})$, we have

$$\widetilde{\mathcal{L}}_{\text{D}_2\text{SM}}(\boldsymbol{\theta}) = \mathbb{E}_{\tilde{\mathbf{x}}, \mathbf{x}, \mathbf{m}} \left[ \left\| \left\{ \mathbf{s}_2(\tilde{\mathbf{x}}; \boldsymbol{\theta}) + \mathbf{s}_1(\tilde{\mathbf{x}})\mathbf{s}_1^\top(\tilde{\mathbf{x}}) + \frac{\mathbf{I} - \mathbf{z}\mathbf{z}^\top}{\sigma^2} \right\} \odot \mathbf{g}_2(\mathbf{x}, \mathbf{m}) \right\|_2^2 \right]$$

$$= \mathbb{E}_{\mathbf{x}, \mathbf{m}} \mathbb{E}_{\mathbf{z} \sim \mathcal{N}(\mathbf{0}, \mathbf{I})} \left[ \left\| \left\{ \boldsymbol{\psi}(\mathbf{x} + \sigma\mathbf{z}; \boldsymbol{\theta}) + \frac{\mathbf{I} - \mathbf{z}\mathbf{z}^\top}{\sigma^2} \right\} \odot \mathbf{g}_2(\mathbf{x}, \mathbf{m}) \right\|_2^2 \right]$$

$$= \mathbb{E}_{\mathbf{x}, \mathbf{m}} \mathbb{E}_{\mathbf{z} \sim \mathcal{N}(\mathbf{0}, \mathbf{I})} \left[ \left\{ \| \boldsymbol{\psi}(\mathbf{x} + \sigma\mathbf{z}; \boldsymbol{\theta}) \|_2^2 + \left\| \frac{\mathbf{I} - \mathbf{z}\mathbf{z}^\top}{\sigma^2} \right\|_2^2 + 2\boldsymbol{\psi}(\mathbf{x} + \sigma\mathbf{z}; \boldsymbol{\theta})\frac{\mathbf{I} - \mathbf{z}\mathbf{z}^\top}{\sigma^2} \right\} \odot \mathbf{g}_2(\mathbf{x}, \mathbf{m}) \right]$$

Denote $\psi_{ij}(\tilde{x}; \theta)$ as the $ij$th term of $\boldsymbol{\psi}(\tilde{\mathbf{x}}; \boldsymbol{\theta})$, $\phi_{ij} = \mathbf{I}_{ij} - \mathbf{z}_i\mathbf{z}_j$ and $g_{ij}$ as the $ij$th term of $\mathbf{g}_2(\mathbf{x}, \mathbf{m})$ and according to Taylor expansion, we have,

$$\mathbb{E}_x \mathbb{E}_{z \sim \mathcal{N}(0, I)} \left[ \left\{ \psi_{ij}(x + \sigma z; \theta)^2 + \frac{\phi_{ij}}{\sigma^2} + 2\psi_{ij}(x + \sigma z; \theta)\frac{\phi_{ij}}{\sigma^2} \right\} \odot g_{ij}(x, m) \right]$$

$$= \mathbb{E}_x \mathbb{E}_{z \sim \mathcal{N}(0, I)} \left[ \left\{ \psi_{ij}(x; \theta)^2 + 2\psi_{ij}(x; \theta)\frac{\phi_{ij}}{\sigma^2} + 2\nabla\psi_{ij}(x; \theta)\frac{\phi_{ij}}{\sigma} + C \right\} \odot g_{ij}(x, m) \right] + \mathcal{O}(1)$$

where $z = \frac{\tilde{x} - x}{\sigma}$, with $C = \left(\frac{\phi_{ij}}{\sigma^2}\right)^2$ and $\nabla\psi_{ij}(x + \sigma z; \theta)$ representing the derivative of $\psi_{ij}(x + \sigma z; \theta)$ with respect to $x$. $\left(\frac{\phi_{ij}}{\sigma^2}\right)^2$ can be treated as a constant that does not depend on $\boldsymbol{\theta}$, and $\mathcal{O}(1)$ remains bounded when $\sigma \to 0$. However, when calculating the expectation from samples, the variances of $\frac{\phi_{ij}}{\sigma^2}$ and $\nabla\psi_{ij}(x; \theta)\frac{\phi_{ij}}{\sigma^2}$ increase without bound as $\sigma \to 0$, due to the presence of $\sigma$ and $\sigma^2$ in the denominator. This causes a significant rise in variance for $\text{D}_2\text{SM}$, making the optimization process more difficult. Consequently, $\text{D}_2\text{SM}$ can become unstable and fail to converge as $\sigma$ approaches zero, necessitating the use of variance reduction techniques.

In this case, we can employ the same variance reduction method outlined in the proof of C.6. However, to bypass the need for estimating $\nabla\psi_{ij}(x; \theta)\frac{\phi_{ij}}{\sigma^2}$, we utilize the antithetic sampling technique same as Meng et al. (2021) to reduce variance.

Denote $\tilde{x}_+ = x + \sigma z$ and $\tilde{x}_- = x + \sigma z$ as the antithetic samples, according to Taylor expansion, the $ij$th term of the $D_2\text{SM}(\theta)$ then becomes,

$$\widetilde{\mathcal{L}}_{D_2\text{SM}}(\theta)_{ij} = \mathbb{E}_x\mathbb{E}_{z\sim\mathcal{N}(0,I)}\left[\left\{\left(\psi_{ij}(\tilde{x}_+;\theta)+\frac{\phi_{ij}}{\sigma^2}\right)^2 + \left(\psi_{ij}(\tilde{x}_-;\theta)+\frac{\phi_{ij}}{\sigma^2}\right)^2\right\}\odot g_{ij}(x,m)\right]$$

$$= \mathbb{E}_x\mathbb{E}_{z\sim\mathcal{N}(0,I)}\left[\left\{\psi_{ij}(\tilde{x}_+;\theta)^2 + 2\frac{\phi_{ij}}{\sigma^2}\psi_{ij}(\tilde{x}_+;\theta) + \psi_{ij}(\tilde{x}_-;\theta)^2 + 2\frac{\phi_{ij}}{\sigma^2}\psi_{ij}(\tilde{x}_-;\theta) + C\right\}\odot g_{ij}(x,m)\right]$$

$$= \mathbb{E}_x\mathbb{E}_{z\sim\mathcal{N}(0,I)}\left[\left\{\psi_{ij}(\tilde{x}_+;\theta)^2 + \psi_{ij}(\tilde{x}_-;\theta)^2 + 2\frac{\phi_{ij}}{\sigma^2}\left(\psi_{ij}(\tilde{x}_+;\theta)+\psi_{ij}(\tilde{x}_-;\theta)\right) + C\right\}\odot g_{ij}(x,m)\right]$$

$$= \mathbb{E}_x\mathbb{E}_{z\sim\mathcal{N}(0,I)}\left[\left\{2\left(\psi_{ij}(x;\theta)\right)^2 + \frac{\phi_{ij}}{\sigma^2}\left[4\psi_{ij}(x;\theta)+2\nabla\psi_{ij}(x;\theta)-2\nabla\psi_{ij}(x;\theta)\right] + C\right\}\odot g_{ij}(x,m)\right]$$

$$= \mathbb{E}_x\mathbb{E}_{\mathbf{z}\sim\mathcal{N}(\mathbf{0},\mathbf{I})}\left[\left\{2\left(\psi_{ij}(x;\theta)\right)^2 + 4\frac{\phi_{ij}}{\sigma^2}\left(\psi_{ij}(x;\theta)\right) + C\right\}\odot g_{ij}(x,m)\right],$$

where $C = 2\left(\frac{\phi_{ij}}{\sigma^2}\right)^2$, a constant with respect to the optimization.

Therefore, we have the variance reduction for $\mathcal{L}_{D_2\text{SM-VR}}(\boldsymbol{\theta})$ which is equivalent to optimizing Eq. (4) up to a control variate. Moreover, when $\sigma$ approaches zero, optimizing Eq. (9) is more stable.

$$\mathcal{L}_{D_2\text{SM-VR}}(\boldsymbol{\theta}) = \mathbb{E}_{\mathbf{x},\mathbf{m}}\mathbb{E}_{\mathbf{z}\sim\mathcal{N}(\mathbf{0},\mathbf{I})}\left[\left\{\boldsymbol{\psi}(\tilde{\mathbf{x}}^+)^2 + \boldsymbol{\psi}(\tilde{\mathbf{x}}^-)^2 + 2\frac{\mathbf{I}-\mathbf{z}\mathbf{z}^\top}{\sigma}\odot\boldsymbol{\Psi}(\cdot)\right\}\odot\mathbf{g}_2(\mathbf{x},\mathbf{m})\right],\tag{18}$$

where the antithetic samples are defined as $\mathbf{x}^+ = \mathbf{x}+\sigma\mathbf{z}$ and $\mathbf{x}^- = \mathbf{x}-\sigma\mathbf{z}$. Here, $\boldsymbol{\psi} = \mathbf{s}_2 + \mathbf{s}_1\mathbf{s}_1^\top$, and $\boldsymbol{\Psi} = \left(\boldsymbol{\psi}(\tilde{\mathbf{x}}^+)+\boldsymbol{\psi}(\tilde{\mathbf{x}}^-)-2\boldsymbol{\psi}(\mathbf{x})\right)$.

# D  DATA GENERATION UNDER DIFFERENT MISSING MECHANISMS

Missing data mechanisms can vary significantly, but they are typically categorized into three main types as defined by (Rubin, 1976): missing completely at random (MCAR), missing at random (MAR), and missing not at random (MNAR). In our experiments, we simulate missing data based on these mechanisms as follows:

**MCAR (Missing Completely at Random)**: Missing values are generated uniformly, with each data point having an equal probability of being missing, determined by a predefined missing rate, $\alpha$. Specifically, missing values are generated using a Bernoulli distribution, $\mathrm{Ber}(\alpha)$, where each entry is missing independently with probability $\alpha$.

**MAR (Missing at Random)**: In this scenario, missing values are generated using a logistic model. A random subset of the variables is selected to remain fully observed, while the remaining variables have missing values depending on the fully observed ones. The missingness is determined by a logistic model with random coefficients, scaled to achieve the target proportion of missing data for the variables influenced by the fully observed subset.

**MNAR (Missing Not at Random)**: The MNAR mechanism is modeled using a logistic masking model. It implements two mechanisms and in either case, weights are random and the intercept is selected to attain the desired proportion of missing values.

- Missing probabilities for each variable are determined by a logistic model that takes all the variables (including those with missing data) as inputs;
- Variables are split into two sets: a set of input variables for the logistic model and a set of variables whose missingness is determined by the logistic model. The input variables are masked using an MCAR process, meaning the missingness in the second set depends on the missingness in the input set.

In all experiments, for MAR missing mechanism, we use logistic regression to estimate the likelihood of each data point being observed. For MNAR, we utilize the same training objective as MCAR, while recognizing that this method may introduce some bias.

The algorithm for training the first- and second-order models under different missing mechanisms is outlined in Algorithm 1:

---

**Algorithm 1** `MissScore`

---

1: **Input:** Observed data $\mathbf{x}_{\mathrm{obs}}$, score models $\mathbf{s}_1(\cdot; \boldsymbol{\theta})$, $\mathbf{s}_2(\cdot; \boldsymbol{\theta})$, noise level $\sigma$, coefficient $\omega$
2: Infer the missingness mask $\mathbf{m} = \mathbf{1}_{[\mathbf{x}_{\mathrm{obs}}=\mathrm{na}]}$
3: **repeat**
4:     Sample noise $\mathbf{z} \sim \mathcal{N}(\mathbf{0}, \mathbf{I})$
5:     Compute perturbed data: $\tilde{\mathbf{x}}_{\mathrm{obs}} = \mathbf{x}_{\mathrm{obs}} + \sigma\mathbf{z}$
6:     **if** missing mechanism is MAR **then**
7:         Estimate $\mathbb{P}[\mathbf{m} = 0|\mathbf{x} = \mathbf{x}_{\mathrm{obs}}]$ and $\mathbb{P}[\mathbf{m}\mathbf{m}^\top = 0|\mathbf{x} = \mathbf{x}_{\mathrm{obs}}]$ using fitted logistic models
8:     **end if**
9:     Update parameters using gradient descent on $\nabla_{\boldsymbol{\theta}}\big(\mathrm{Eq.}(5) + \omega\,\mathrm{Eq.}(6)\big)$
10: **until** convergence

---

# E  ADDITIONAL INFORMATION ON SAMPLING

In the sampling experiments with the Swiss-Roll dataset under MCAR, we use a small perturbation $\sigma = 0.01$ and jointly optimize Eq. (8) and Eq.(9), where $\mathbf{s}_1(\tilde{\mathbf{x}}) \approx \mathbf{s}_1(\mathbf{x})$ and $\mathbf{s}_2(\tilde{\mathbf{x}}) \approx \mathbf{s}_2(\mathbf{x})$. The sample size is set to 5000. Both $\mathbf{s}_1(\tilde{\mathbf{x}}; \boldsymbol{\theta})$ and $\mathbf{s}_2(\tilde{\mathbf{x}}; \boldsymbol{\theta})$ are modeled using a 3-layer MLP with a latent size of 128 and a Softplus activation function. We use a learning rate of 0.001, a batch size of 64 and train for 100 epochs, which takes approximately 4 minutes on an Intel(R) Xeon(R) Gold 6448H CPU. The experiments in Section 3 also utilize the same model configuration for training. In the Ozaki sampling experiments, we only use the diagonal of $\mathbf{s}_2(\tilde{\mathbf{x}}; \boldsymbol{\theta})$ to avoid the computational costs associated with the inversion, exponentiation, and decomposition of $\mathbf{s}_2(\tilde{\mathbf{x}}; \boldsymbol{\theta})$. The algorithm is presented in Algorithm 2.

---

**Algorithm 2** `MissScore-Sampling`

---

1: **Input:** Score models $\mathbf{s}_1(\cdot; \boldsymbol{\theta}^*)$, $\mathbf{s}_2(\cdot; \boldsymbol{\theta}^*)$; step size $\epsilon$; number of iterations $T$
2: **Initialize:** $\tilde{\mathbf{x}}_0 \sim \pi(\mathbf{x})$
3: **for** $t = 1$ to $T$ **do**
4:     Sample noise $\mathbf{z}_t \sim \mathcal{N}(\mathbf{0}, \mathbf{I})$
5:     **if** Ozaki sampling **then**
6:         Compute $\mathbf{M}_{t-1} = \left(e^{\epsilon \mathbf{s}_2(\tilde{\mathbf{x}}_{t-1})} - \mathbf{I}\right)\mathbf{s}_2(\tilde{\mathbf{x}}_{t-1})^{-1}$
7:         Compute $\Sigma_{t-1} = \left(e^{2\epsilon \tilde{\mathbf{x}}_{t-1}} - \mathbf{I}\right)\mathbf{s}_2(\tilde{\mathbf{x}}_{t-1})^{-1}$
8:         Update $\tilde{\mathbf{x}}_t = \tilde{\mathbf{x}}_{t-1} + \mathbf{M}_{t-1}\mathbf{s}_1(\tilde{\mathbf{x}}_{t-1}) + \Sigma_{t-1}^{1/2}\mathbf{z}_t$
9:     **else**
10:         Update $\tilde{\mathbf{x}}_t = \tilde{\mathbf{x}}_{t-1} + \frac{1}{2}\epsilon \mathbf{s}_1(\tilde{\mathbf{x}}_{t-1}) + \sqrt{\epsilon}\mathbf{z}_t$
11:     **end if**
12: **end for**
13: **Return:** $\tilde{\mathbf{x}}_T$

---

The following sections provide experimental details on data generation with the simulated Bayesian Network and real Census data.

## E.1  DATASET DESCRIPTION AND PROCESSING

**Bayesian Network** Details regarding the data generated from a Bayesian Network can be found in Section B.1 in Ouyang et al. (2023).

**Census** Census dataset is a binary classification dataset that predict whether income exceeds 50K/yr based on census data (Kohavi, 1996). Also known as Adult dataset.

The statistical information of datasets used in our experiments is in Table 5. #train, #test, #continuous, and #categorical mean the number of training data, testing data, continuous columns, and categorical columns, respectively.

Table 5: Synthetic and Real-World Datasets Used in Experiments.

| Dataset | #Train | #Test | #Categorical | #Continuous |
|---|---|---|---|---|
| Bayesian Network | 2000 | 20000 | 3 | 2 |
| Census | 16000 | 4000 | 9 | 6 |

For data processing, we follow standard pre- and post-processing procedures for mixed-type tabular data. Specifically, we apply min-max normalization to continuous variables and reverse this scaling during generation. For discrete variables, we use one-hot encoding and apply a rounding function after the softmax function during generation.

## E.2 Evaluation Methods

We adopt the "train on synthetic, test on real (TSTR)" framework (Esteban et al., 2017), a widely used method for assessing the quality of sampling data from generative model (Kim et al., 2022; Ouyang et al., 2023; Li et al., 2019). The experimental results for sampling in this paper are calculated as follows:

1. We first download a dataset and use its existing train-test split.

2. Then we generate synthetic records equal in number to the original training set using various synthetic data generation methods.

3. Using the synthetic training records from Step 2, we train base classifiers to make predictions. We conduct a hyperparameter search for each classifier, considering Decision Tree, AdaBoost, Logistic Regression, MLP Classifiers, Random Forest, and XGBoost for the classification tasks. The hyperparameters and their candidate settings follow those described in Kim et al. (2022); Ouyang et al. (2023), and are summarized in Table 26 of Kim et al. (2022).

4. Finally, we evaluate the classifiers using the testing dataset, applying a range of evaluation metrics for comprehensive assessment.

Steps 2 to 4 are repeated three times for each dataset, and the average scores for each method across all evaluation metrics are calculated. The detailed metrics used in our experiment include:

1. **Accuracy**: This is calculated using the `accuracy_score` function from the `sklearn.metrics` module.

2. **Weighted-F1**:

$$\text{Weighted-F1} = \sum_{i=0}^{N} w_i s_i$$

   where $N$ is the total number of classes. The weight for the $i$-th class, $w_i = \frac{1-p_i}{N-1}$, with $p_i$ representing the proportion of the $i$-th class's size relative to the total dataset. Here, $s_i$ is the F1 score for the $i$-th class, calculated using the One-vs-Rest strategy. This weighting approach is designed to prioritize the evaluation of synthesized tables by giving more importance to smaller classes, which are often prone to being overlooked by the model, thus addressing mode collapse.

3. **AUROC**: This is calculated using the `roc_auc_score` function from the `sklearn.metrics` module.

4. **SDMetrics**: This metric evaluates synthetic data by comparing it against the real data, as described in (Dat, 2023).

Among all metrics, a higher score indicates better overall quality of the synthetic data.

## E.3 Model Architecture

We use a perturbation of $\sigma = 0.1$ and jointly optimize Eq. (8) and Eq. (9). In the Bayesian Network experiment, we follow the same configuration as described earlier, with each run taking approximately 20 minutes. For the census dataset, we employ a simple MLP consisting of 5 Linear layers, LeakyReLU activation, Layer Normalization, and Dropout with a probability of 0.2 in the first layer. The learning rate is set to 0.001. The first two layers use a latent size of 128, while the last three layers use a latent size of 1024. We train with a batch size of 256 for 250 epochs, with each experiment taking approximately 4 hours. All experiments are performed on an Intel(R) Xeon(R) Gold 6448H CPU. For the downstream classifier, we use the same base hyperparameters as listed in Table 26 of Kim et al. (2022).

## E.4 Experimental Results

In the following experimental results, we use a missing data ratio of $\alpha = 0.3$ and apply XGBoost for the downstream tasks, without delving into specific implementation details. Table 6 presents the

utility evaluation of MissScore using both Langevin and Ozaki samplings, compared to other baseline methods, on the Census dataset with a missing ratio of 0.3. Additionally, Table 7 summarizes the Accuracy, AUROC, and Weighted-F1 metrics as the missing ratio varies. Figure 5 illustrates the fidelity evaluation of MissScore, again using Langevin and Ozaki samplings alongside other baselines, on the Bayesian dataset.

Table 6: Utility evaluation of MissScore using Langevin and Ozaki samplings, along with other baselines, on the Census dataset with missing ratio 0.3.

| Criterion | Mechanism | Langevin | Ozaki | DSM-delete | DSM-mean | STaSy-mean |
|---|---|---|---|---|---|---|
| | MCAR | 0.80 | **0.81** | 0.70 | 0.75 | 0.77 |
| Accuracy | MAR | **0.82** | **0.82** | 0.69 | 0.77 | 0.74 |
| | MNAR | **0.81** | 0.80 | 0.59 | 0.80 | 0.75 |
| | MCAR | **0.84** | 0.84 | 0.57 | 0.67 | 0.62 |
| AUROC | MAR | 0.85 | **0.86** | 0.46 | 0.75 | 0.61 |
| | MNAR | **0.86** | **0.86** | 0.52 | 0.76 | 0.63 |
| | MCAR | **0.52** | **0.52** | 0.24 | 0.32 | 0.41 |
| Weighted-F1 | MAR | **0.61** | 0.60 | 0.32 | 0.38 | 0.38 |
| | MNAR | **0.69** | 0.68 | 0.41 | 0.52 | 0.42 |

Table 7: Evaluation of MissScore using Ozaki samplings on the Census dataset with varying missing ratios $\{0.1, 0.3, 0.5, 0.7, 0.9\}$.

| Methods | | $\alpha = 0.1$ | $\alpha = 0.3$ | $\alpha = 0.5$ | $\alpha = 0.7$ | $\alpha = 0.9$ |
|---|---|---|---|---|---|---|
| | MCAR | 0.81 | 0.81 | 0.80 | 0.79 | 0.77 |
| Accuracy | MAR | 0.71 | 0.82 | 0.82 | 0.82 | 0.79 |
| | MNAR | 0.80 | 0.80 | 0.83 | 0.74 | 0.72 |
| | MCAR | 0.85 | 0.84 | 0.84 | 0.86 | 0.61 |
| AUROC | MAR | 0.85 | 0.86 | 0.87 | 0.85 | 0.83 |
| | MNAR | 0.85 | 0.87 | 0.86 | 0.85 | 0.80 |
| | MCAR | 0.54 | 0.52 | 0.41 | 0.66 | 0.22 |
| Weighted-F1 | MAR | 0.64 | 0.60 | 0.67 | 0.65 | 0.63 |
| | MNAR | 0.46 | 0.68 | 0.61 | 0.64 | 0.63 |

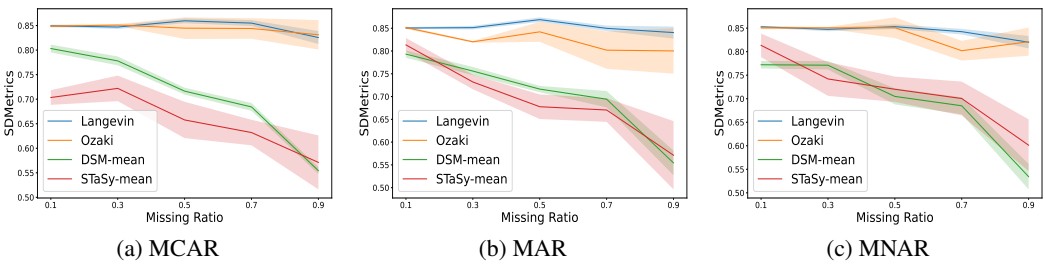

(a) MCAR        (b) MAR        (c) MNAR

Figure 5: Fidelity evaluation of MissScore using Langevin and Ozaki samplings, along with other baselines, on the Census dataset varies with missing ratio $\alpha = \{0.1, 0.3, 0.5, 0.7\}$ under different missing mechanisms.

E.5   ANALYSIS OF SAMPLING RESULTS USING CENSUS DATA

In this section, we analyze the sampling performance using the first- and second-order models at a missing ratio of 0.3 with real Census data. Our analysis consists of two parts: numerical and

categorical data in the census dataset. For the numerical data, which includes variables such as age, final weight (fnlwgt), years of education completed (education-num), and hours worked per week, we examine the distribution of both the original data and the sampled data shown in Figures 6 to 9. This comparison is done using MissScore with first-order information via Langevin dynamics (equivalent to MissDiff) and MissScore with second-order information through Ozaki sampling. We observe that MissScore with second-order information captures finer details in the distribution. For instance, in the age range of 20-40, the fnlwgt range of 0-0.25, education-num range of 10-11, and hours per week range of 40-45, the second-order MissScore better approximates the shape of the original distribution.

For the categorical data, we analyze the education variable and again find that the second-order MissScore aligns more closely with the original distribution, as shown in Figure 10. These results indicate that MissScore with second-order information captures finer-grained details in the sampled dataset, compared to MissDiff (MissScore with first-order information).

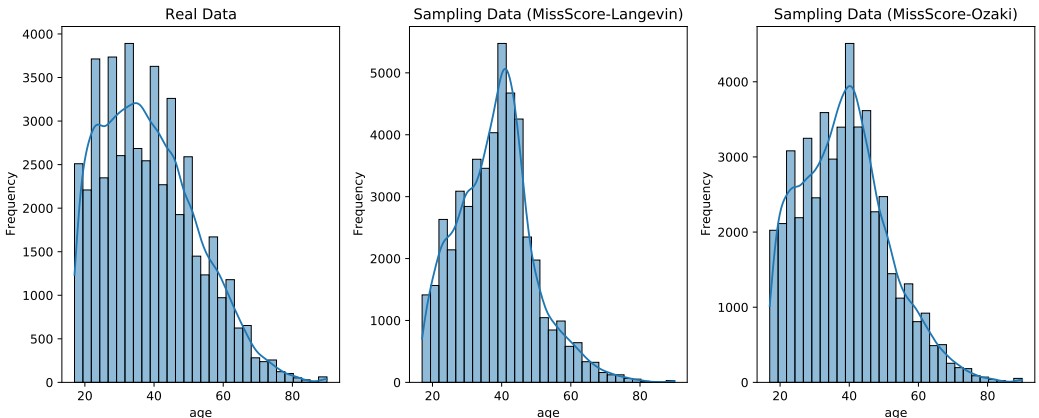

Figure 6: Analysis of Age in the Census Dataset: The first plot represents the actual data, the second plot shows the sampled data using MissScore with Langevin dynamics, and the third plot displays the sampled data using MissScore with Ozaki sampling.

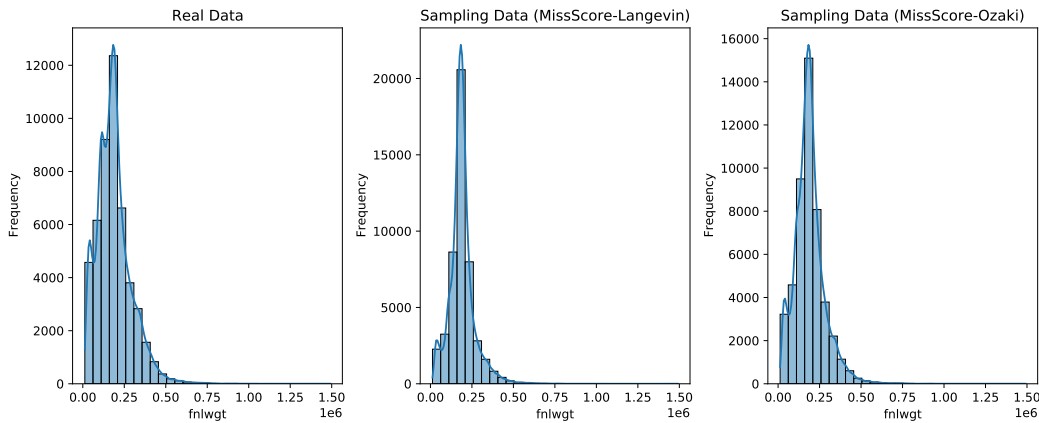

Figure 7: Analysis of Final Weights in the Census Dataset: The first plot represents the actual data, the second plot shows the sampled data using MissScore with Langevin dynamics, and the third plot displays the sampled data using MissScore with Ozaki sampling.

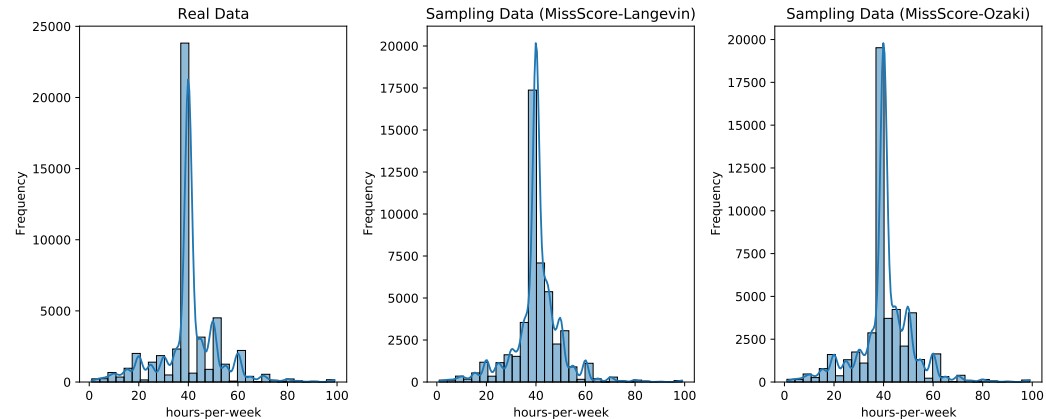

Figure 9: Analysis of Hours-per-week in the Census Dataset: The first plot represents the actual data, the second plot shows the sampled data using MissScore with Langevin dynamics, and the third plot displays the sampled data using MissScore with Ozaki sampling.

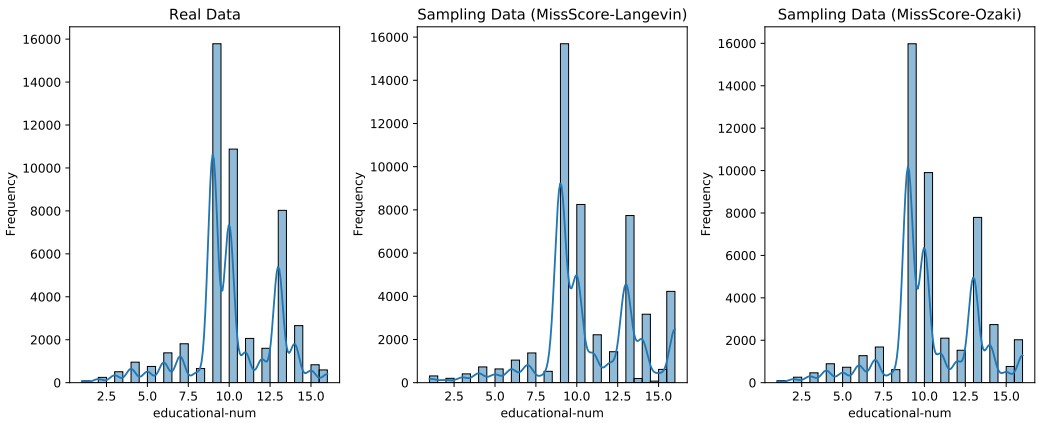

Figure 8: Analysis of The number of years of education completed in the Census Dataset: The first plot represents the actual data, the second plot shows the sampled data using MissScore with Langevin dynamics, and the third plot displays the sampled data using MissScore with Ozaki sampling.

## F ADDITIONAL INFORMATION ON CAUSAL DISCOVERY

### F.1 RELATED WORK

**Causal Discovery with Complete data.** Causal discovery aims to uncover the underlying causal relationships among variables of interest from purely observational data, specifically identifying a causal Directed Acyclic Graph (DAG) for a given dataset. This problem lies at the heart of causal inference, as knowledge of the causal graph enables prediction of the effects of interventions. However, causal discovery from observational data is inherently ill-posed, necessitating additional assumptions, such as imposing functional assumptions on the data-generating process. We adopt the notion of structural causal model (SCM) to characterize the causal relations among variables. Each SCM $\mathcal{M} = \langle \mathcal{Z}, \mathcal{X}, \mathcal{F} \rangle$ consists of the exogenous variable set $\mathcal{Z} = \{Z_1, Z_2, \ldots, Z_d\}$, the endogenous variable set $\mathcal{X} = \{X_1, X_2, \ldots, X_d\}$, and the function set $\mathcal{F} = \{f_1, f_2, \ldots, f_d\}$. Here, each function $f_i$ computes the variable $X_i$ from its parents (or causes) $X_{\mathrm{PA}_i}$ and an exogenous variable $Z_i$, i.e., $X_i = f_i(X_{\mathrm{PA}_i}, Z_i)$. We focus on a specific class of SCMs, called the additive noise models

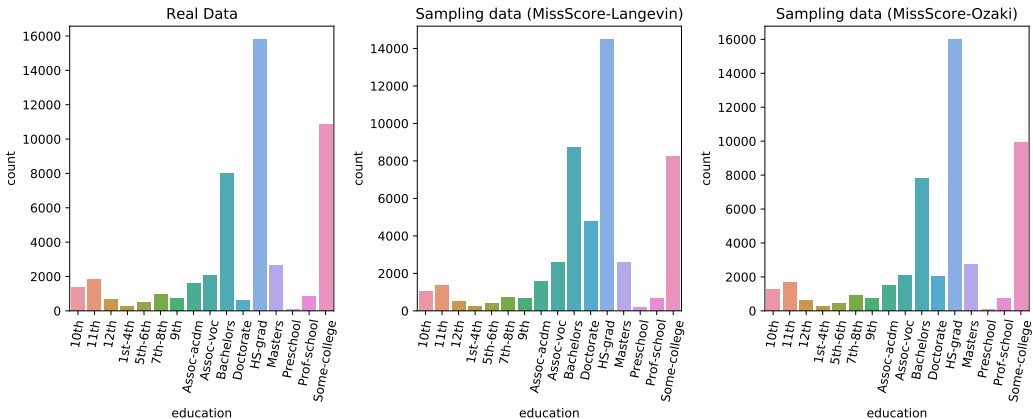

Figure 10: Analysis of Education in the Census Dataset: The first plot represents the actual data, the second plot shows the sampled data using MissScore with Langevin dynamics, and the third plot displays the sampled data using MissScore with Ozaki sampling.

(ANMs), given by $X_i = f_i(X_{\mathrm{PA}_i}) + Z_i, \quad i = 1, 2, \ldots, d$, where $Z_i$, interpreted as the additive noise variable, is assumed to be independent of variables in $X_{\mathrm{PA}_i}$ and mutually independent with variables in $\mathcal{Z} \setminus Z_i$.

Rolland et al. (2022) proposed an order-based algorithm for this model, further assuming that $f_i$ is a twice-differentiable nonlinear function and $Z_i$ is Gaussian noise. This method enables the identification of leaf nodes based on the diagonal of the Hessian of the log-likelihood. Before proceeding to the method for identifying leaves, we first derive an analytical expression for the score following Lemma 2 in Rolland et al. (2022). The score is written as follows:

$$\nabla_{x_j} \log p(\mathbf{x}) = \nabla_{x_j} \log \prod_{i=1}^{d} p(x_i \mid x_{\mathrm{PA}_i})$$

$$= \nabla_{x_j} \sum_{i=1}^{d} \log p(x_i \mid x_{\mathrm{PA}_i})$$

$$= \nabla_{x_j} \sum_{i=1}^{d} \log p(x_i - f_i) \quad \text{(where } z_i = x_i - f_i(x_{\mathrm{PA}_i}))$$

$$= \frac{\partial \log p(x_j - f_j(x_{\mathrm{PA}_j}))}{\partial x_j} - \sum_{i \in \mathrm{CH}_j} \frac{\partial f_i}{\partial x_j} \frac{\partial \log p(x_i - f_i(x_{\mathrm{PA}_i}))}{\partial x}.$$

where $\mathrm{CH}_j$ represents the children of the variable $j$. As a result, $\frac{\partial}{\partial x_j} \nabla_{x_j} \log p(\mathbf{x}) = a$, where $a$ is a constant, Consequently, the variance of the diagonal elements of the Hessian is zero (i.e. $\mathrm{Var}_{\mathbf{X}}[H_{j,j}(\log p(\mathbf{x}))] = 0$) if and only if node $j$ is a leaf node.

However, existing computational methods for calculating the Hessian struggle to scale efficiently as the number of variables and samples increases, limiting the scalability of Rolland et al. (2022). To address this, Sanchez et al. (2022) introduced a diffusion-based model that efficiently computes the Hessian, enabling the method to scale to larger datasets, both in terms of sample size and number of variables, while maintaining comparable performance to Rolland et al. (2022).

**Causal Discovery with Incomplete Data.** Several extensions of the PC algorithm have been developed to learn causal graphs from incomplete data (Tu et al., 2019; Gain & Shpitser, 2018), utilizing only the fully observed samples while mitigating biases in conditional independence tests. Another prominent family of methods relies on Expectation-Maximization (Dempster et al., 1977), where missing values are iteratively inferred while simultaneously learning the causal structure. Building on the continuous optimization techniques introduced by NOTEARS (Zheng et al., 2018), MissDAG (Gao et al., 2022) extends this approach to continuous identifiable Additive Noise Models (ANMs),

using approximate posterior inference via Monte Carlo and rejection sampling when the exact posterior is unavailable. MissOTM (Vo et al., 2024) introduces a score-based method that leverages optimal transport to learn causal structures from incomplete data. A distinct approach is taken by VISL (Morales-Alvarez et al., 2022), which employs amortized variational inference in a Bayesian framework. Unlike MissOTM and MissDAG, VISL assumes a latent low-dimensional factor that captures the essential structure of the data based on observed variables. The latent factors are then used to reconstruct the complete data and discover the underlying causal graph.

### F.2 EVALUATION METRICS

For each method, we compute the

**SHD.** Structural Hamming distance between the output and the true causal graph, which counts the number of missing, falsely detected, or reversed edges.

**Order Divergence.** Rolland et al. (2022) propose this quantity for measuring how well the topological order is estimated. For an ordering $\pi$, and a target adjacency matrix A, we define the topological order divergence $D_{\text{top}}(\pi, \mathbf{A})$ as

$$D_{\text{top}}(\pi, \mathbf{A}) = \sum_{i=1}^{d} \sum_{j:\pi_i > \pi_j} \mathbf{A}_{ij} \tag{19}$$

### F.3 MODEL ARCHITECTURE

We apply a perturbation of $\sigma = 0.1$ and jointly optimize Eq. (8) and Eq. (9). The model is a simple MLP with 5 Linear layers, LeakyReLU activation, Layer Normalization, and a Dropout rate of 0.2 in the first layer. The learning rate is set to 0.001. The first two layers have a latent size of $\max(128, 3 \times d)$, while the last three use a latent size of $\max(1024, 5 \times d)$. Training is conducted with a batch size of 128 for 150 epochs. The time efficiency is shown in the figure 4 with ER graph model across various dimensions. All experiments are executed on an Intel(R) Xeon(R) Gold 6448H CPU. The algorithm is presented in Algorithm 3.

---

**Algorithm 3** `MissScore-Causal Discovery`

---

1: **Input:** Observed data $\mathbf{x}_{\text{obs}}$; score models $\mathbf{s}_1(\cdot; \boldsymbol{\theta})$, $\mathbf{s}_2(\cdot; \boldsymbol{\theta})$
2: **Initialize:** $\pi = []$; nodes $= \{1, \dots, d\}$
3: $n, d \leftarrow$ shape of $\mathbf{x}_{\text{obs}}$
4: **for** $k = 1$ to $d$ **do**
5:     Jointly train the score models $\mathbf{s}_1(\boldsymbol{\theta})$ and $\mathbf{s}_2(\boldsymbol{\theta})$ using $\mathbf{x}_{\text{obs}}$ with Algorithm 1
6:     Generate $n$ samples $\tilde{\mathbf{x}}_{\text{new}}$ using Algorithm 2 with bootstrapping
7:     Estimate the second-order score $\mathbf{s}_2(\tilde{\mathbf{x}}_{\text{new}})$ using $\mathbf{s}_2(\boldsymbol{\theta})$
8:     $V_j = \text{Var}_X[\text{diag}(\mathbf{s}_2(\tilde{\mathbf{x}}_{\text{new}}))]$
9:     $\ell \leftarrow \arg\min_{j \in \text{nodes}} V_j$                    ▷ The leaf node
10:     $\pi \leftarrow [\ell, \pi]$                    ▷ Update topological order
11:     nodes $\leftarrow$ nodes $- \{\ell\}$                    ▷ Remove node $\ell$
12:     Remove the $\ell$-th column from $\mathbf{x}_{\text{obs}}$
13: **end for**
14: Obtain the final DAG using CAM pruning associated with the topological order $\pi$.

---

### F.4 EXPERIMENTAL RESULTS

We begin by conducting experiments using complete data, evaluating our approach alongside various missing mechanisms and different missing ratios. Additionally, we include order divergence in the Table. Our findings reveal that, with complete data, MissScore performs on par with DiffAN, as illustrated in Figure 11. Notably, among all settings, MissScore achieves performance similar

to current state-of-the-art approaches while offering superior computational efficiency in terms of memory and time. This scalability is a key advantage where other methods may struggle.

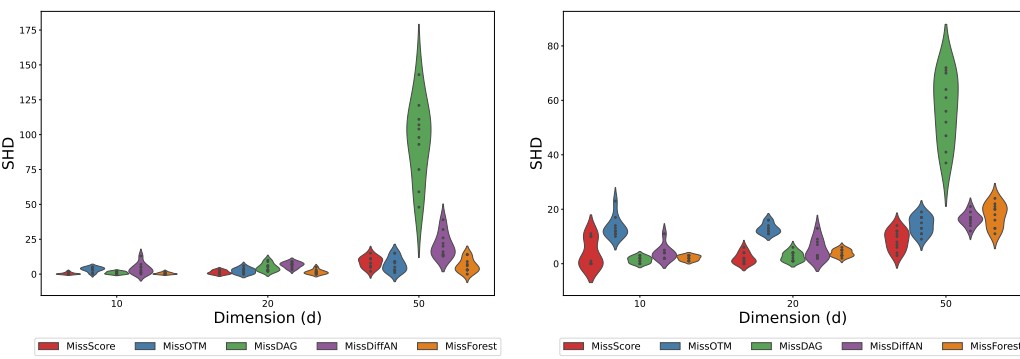

Figure 11: The data is generated using an ER graph model with different dimensions $d = \{10, 20, 50\}$ and an equal number of edges. Each dataset consists of 1000 samples. Left: $f_i$ is an MLP; Right: $f_i$ corresponds to MIM.

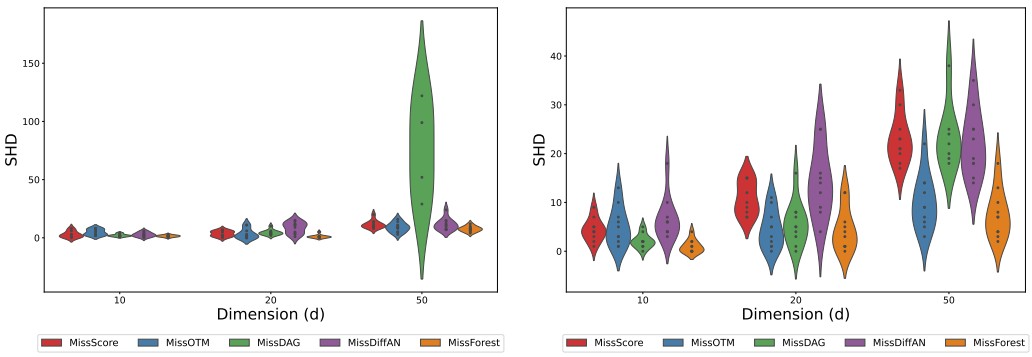

Figure 12: The data is generated under MCAR with missing ratios of 0.1 and 0.3, using an ER graph model with different dimensions $d = \{10, 20, 50\}$ and an equal number of edges. Each dataset consists of 1000 samples. $f_i$ corresponds to MLP. Left: SHD with missing ratio 0.1; Right: SHD with missing ratio 0.3.

Table 8: Order divergence with missing ratios of $\alpha = \{0.1, 0.3\}$ across different missing data mechanisms. The ER graph model is considered with varying dimensions $d = \{10, 20, 50\}$, and an equal number of edges. Each dataset consists of 1000 samples and $f_i$ corresponds to MLP. Lower order divergence indicating better performance.

| Dimensions | Methods | MCAR | | MAR | | MNAR | |
|---|---|---|---|---|---|---|---|
| | | $\alpha = 0.1$ | $\alpha = 0.3$ | $\alpha = 0.1$ | $\alpha = 0.3$ | $\alpha = 0.1$ | $\alpha = 0.3$ |
| d=10 | MissScore | $1.62 \pm 0.99$ | $1.60 \pm 0.66$ | $1.33 \pm 0.47$ | $1.89 \pm 0.75$ | $1.33 \pm 0.47$ | $1.44 \pm 0.50$ |
| | MissDiffAN | $2.00 \pm 1.18$ | $2.90 \pm 1.87$ | $2.75 \pm 1.30$ | $3.60 \pm 1.28$ | $2.60 \pm 1.20$ | $2.30 \pm 1.35$ |
| d=20 | MissScore | $2.10 \pm 0.94$ | $2.67 \pm 2.00$ | $2.70 \pm 1.85$ | $1.94 \pm 0.29$ | $1.70 \pm 1.78$ | $0.70 \pm 0.46$ |
| | MissDiffAN | $4.30 \pm 2.00$ | $4.22 \pm 2.35$ | $4.33 \pm 2.11$ | $3.00 \pm 1.00$ | $2.00 \pm 0.89$ | $3.40 \pm 1.20$ |
| d=50 | MissScore | $3.40 \pm 2.33$ | $4.00 \pm 3.10$ | $3.10 \pm 1.70$ | $4.60 \pm 2.91$ | $2.80 \pm 0.98$ | $4.10 \pm 3.30$ |
| | MissDiffAN | $4.50 \pm 2.91$ | $3.60 \pm 3.32$ | $8.33 \pm 3.65$ | $7.40 \pm 2.84$ | $3.50 \pm 1.75$ | $3.20 \pm 3.28$ |

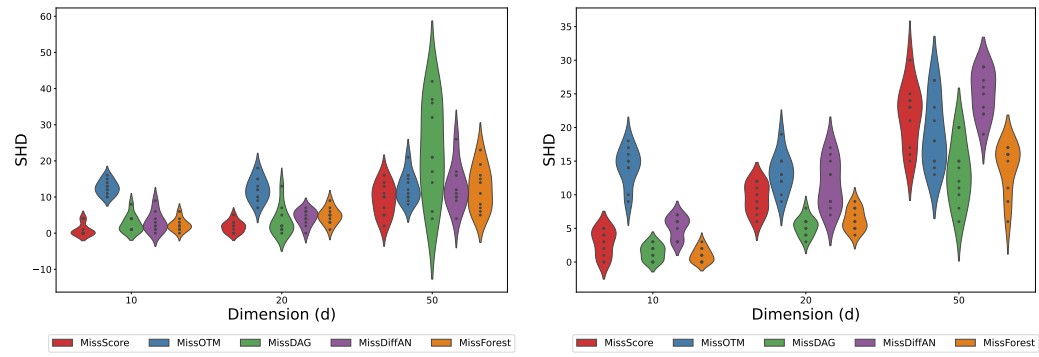

Figure 13: The data is generated under MCAR with missing ratios of 0.1 and 0.3, using an ER graph model with different dimensions $d = \{10, 20, 50\}$ and an equal number of edges. Each dataset consists of 1000 samples. $f_i$ corresponds to MIM. Left: SHD with missing ratio 0.1; Right: SHD with missing ratio 0.3.

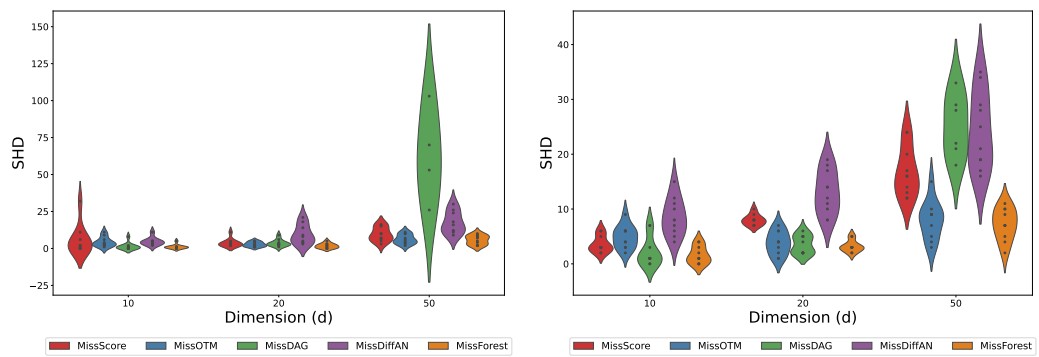

Figure 14: The data is generated under MAR with missing ratios of 0.1 and 0.3, using an ER graph model with different dimensions $d = \{10, 20, 50\}$ and an equal number of edges. Each dataset consists of 1000 samples. $f_i$ corresponds to MLP. Left: SHD with missing ratio 0.1; Right: SHD with missing ratio 0.3.

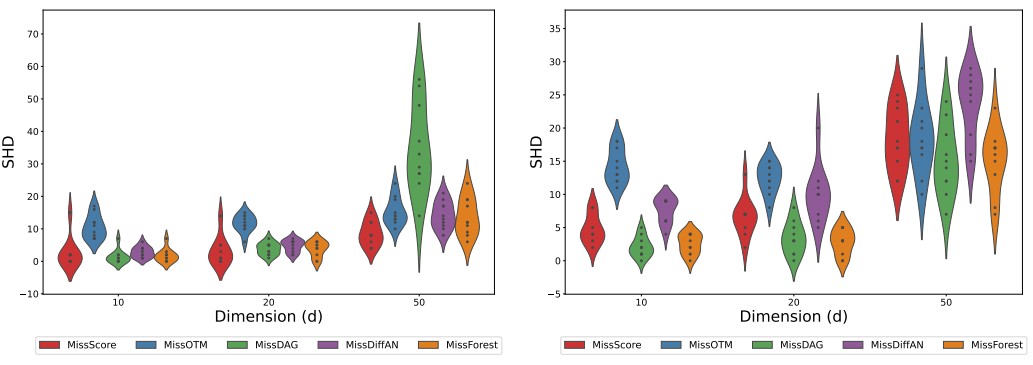

Figure 15: The data is generated under MAR with missing ratios of 0.1 and 0.3, using an ER graph model with different dimensions $d = \{10, 20, 50\}$ and an equal number of edges. Each dataset consists of 1000 samples. $f_i$ corresponds to MIM. Left: SHD with missing ratio 0.1; Right: SHD with missing ratio 0.3.

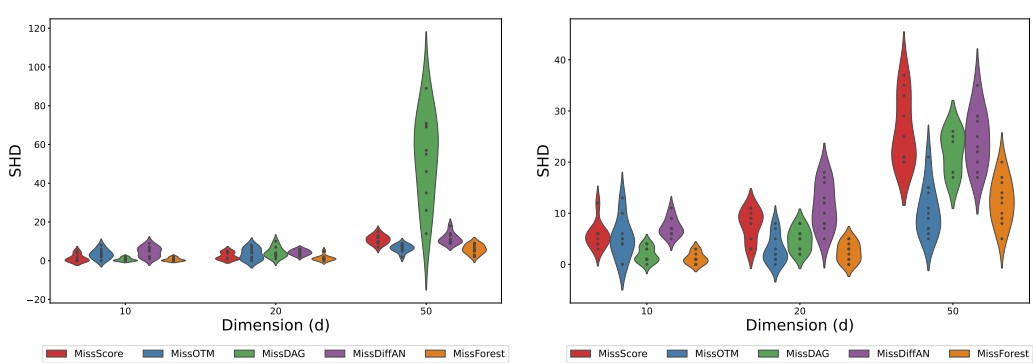

Figure 16: The data is generated under MNAR with missing ratios of 0.1 and 0.3, using an ER graph model with different dimensions $d = \{10, 20, 50\}$ and an equal number of edges. Each dataset consists of 1000 samples. $f_i$ corresponds to MLP. Left: SHD with missing ratio 0.1; Right: SHD with missing ratio 0.3.

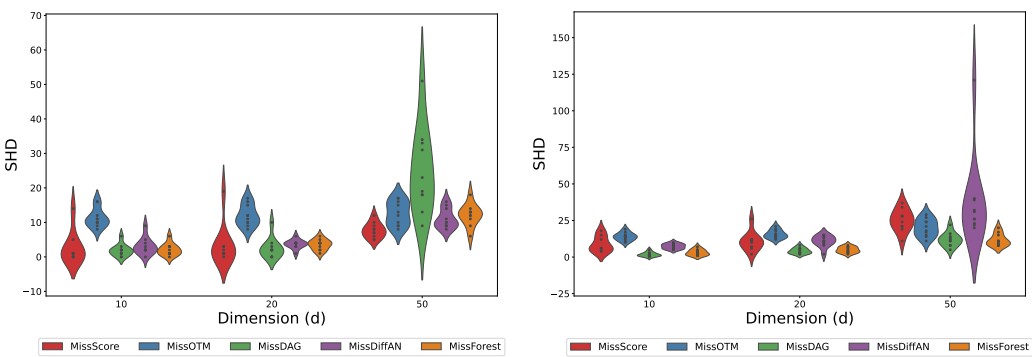

Figure 17: The data is generated under MNAR with missing ratios of 0.1 and 0.3, using an ER graph model with different dimensions $d = \{10, 20, 50\}$ and an equal number of edges. Each dataset consists of 1000 samples. $f_i$ corresponds to MIM. Left: SHD with missing ratio 0.1; Right: SHD with missing ratio 0.3.

Table 9: Order divergence with missing ratios of $\alpha = \{0.1, 0.3\}$ across different missing data mechanisms. The ER graph model is considered with varying dimensions $d = \{10, 20, 50\}$, and an equal number of edges. Each dataset consists of 1000 samples and $f_i$ corresponds to MIM. Lower order divergence indicating better performance.

| Dimensions | Methods | MCAR | | MAR | | MNAR | |
|---|---|---|---|---|---|---|---|
| | | $\alpha = 0.1$ | $\alpha = 0.3$ | $\alpha = 0.1$ | $\alpha = 0.3$ | $\alpha = 0.1$ | $\alpha = 0.3$ |
| d=10 | MissScore | $1.60 \pm 1.20$ | $1.20 \pm 0.60$ | $1.35 \pm 0.47$ | $1.88 \pm 0.93$ | $1.62 \pm 0.99$ | $1.38 \pm 0.48$ |
| | MissDiffAN | $2.22 \pm 2.10$ | $1.50 \pm 0.67$ | $2.50 \pm 1.36$ | $3.70 \pm 1.10$ | $2.40 \pm 2.20$ | $2.20 \pm 1.40$ |
| d=20 | MissScore | $1.70 \pm 0.90$ | $2.20 \pm 1.33$ | $1.82 \pm 1.29$ | $1.90 \pm 1.30$ | $1.56 \pm 0.68$ | $0.56 \pm 0.83$ |
| | MissDiffAN | $2.70 \pm 1.49$ | $3.8 \pm 1.47$ | $3.70 \pm 1.35$ | $3.10 \pm 2.21$ | $2.44 \pm 1.07$ | $2.70 \pm 1.68$ |
| d=50 | MissScore | $4.90 \pm 2.12$ | $4.00 \pm 1.41$ | $4.56 \pm 2.27$ | $4.60 \pm 2.24$ | $4.20 \pm 1.66$ | $4.40 \pm 1.85$ |
| | MissDiffAN | $5.56 \pm 2.45$ | $4.40 \pm 1.56$ | $7.90 \pm 2.39$ | $7.50 \pm 2.69$ | $5.00 \pm 1.61$ | $6.20 \pm 2.68$ |

