# OpenReview forum: "MissScore: High-Order Score Estimation in the Presence of Missing Data"
_ICLR.cc/2025/Conference — Submitted to ICLR 2025_

### Official Review · Reviewer_bLh8 · 2024-10-27

**Soundness:** 3
**Presentation:** 3
**Contribution:** 3
**Rating:** 6
**Confidence:** 3

**Summary:**

The authors present MissScore, a score-based framework for learning high-order scores directly from incomplete data without imputation. They derive objective functions for different missing data mechanisms and demonstrate the framework's effectiveness in approximating second-order scores. They show MissScore improves sampling speed and data quality in generation tasks with missing data, and propose a new causal discovery method.

**Strengths:**

1. The proposed approach addresses limitations in other models (e.g., GANs or VAEs) that fail to estimate high-order scores naturally, especially under missing data conditions. By deriving objective functions suited for missing data mechanisms and focusing on second-order scores, MissScore can improve accuracy and efficiency in approximating high-order scores.

2. The paper proposes a causal discovery method that leverages MissScore’s second-order score model, effectively handling incomplete data. Unlike traditional imputation-based methods, this model integrates high-order score information directly, preserving data structure and reducing potential inaccuracies.

**Weaknesses:**

• The authors' claim that "imputation methods can compromise data quality, potentially leading to biased results and significantly degrading performance in downstream tasks" (also repeated in the Conclusion and Appendix A) would benefit from clarification. Specifically, the cited Ouyang (2023) paper focuses on an **"impute-then-generate"** pipeline where imputation may reduce generation quality by introducing biases. However, **"impute-then-predict"** pipelines often show improved predictive accuracy for downstream classification and regression tasks, as established in works such as:

    •  Poulos, J. and Valle, R. (2018). Missing data imputation for supervised learning. Applied Artificial Intelligence 32, 186–196
    • Jäger, Sebastian, Arndt Allhorn, and Felix Bießmann. "A benchmark for data imputation methods." Frontiers in Big Data 4 (2021)
    • Perez-Lebel, Alexandre, et al. "Benchmarking missing-values approaches for predictive models on health databases." GigaScience 11 (2022)
    • Shadbahr, T., Roberts, M., Stanczuk, J. et al. The impact of imputation quality on machine learning classifiers for datasets with missing values. Commun Med 3, 139 (2023).
    • Paterakis, George, et al. "Do we really need imputation in AutoML predictive modeling?." ACM Transactions on Knowledge Discovery from Data 18.6 (2024): 1-64.

• To address the statement on imputation methods not accounting for uncertainty in missing data, it would benefit the authors to include multiple imputation (MI) as an alternative that does incorporate uncertainty in imputed values. The work of Van Buuren & Groothuis-
Oudshoorn (2011), which introduces a popular MI algorithm (MICE), is cited in Appendix A but not contextualized. MI can be applied with deep learning models like GAIN and MIDA to generate multiple plausible imputed datasets by rerunning these models with varying random initializations, thus capturing uncertainty. Wang et al. (2022) compare MI methods, including deep learning approaches like GAIN and MIDA, finding that MICE with classification trees often outperform deep learning methods in terms of imputation quality, especially in survey data.

    • Gondara, L., & Wang, K. (2018). "MIDA: Multiple imputation using denoising autoencoders." *Advances in Knowledge Discovery and Data Mining: PAKDD*.
    • Wang, Z., Akande, O., Poulos, J., & Li, F. (2022). "Are deep learning models superior for missing data imputation in surveys? Evidence from an empirical comparison." *Survey Methodology*, 48(2), 375-399.

• The differences between MissScore and MissDiff are not entirely clear. To clearly differentiate their contributions, the authors could explicitly state that, unlike MissDiff, which focuses on a first-order diffusion-based approach for data generation, MissScore introduces a framework for learning high-order scores from incomplete data without imputation, specifically targeting the Hessian to enhance downstream performance. The authors should emphasize that MissScore integrates a new causal discovery method, contrasting with MissDiff's focus on unbiased data synthesis.

• To strengthen the conclusion, the authors could acknowledge a few limitations of MissScore. While MissScore performs efficiently in high-dimensional settings, its accuracy in second-order score estimation (measured by SHD) slightly declines as dimensionality increases, particularly for complex causal structures (e.g., d = 50 in Figure 4). While the model maintains high fidelity and utility with moderate missing ratios (\alpha = 0.1 or \alpha = 0.3), its performance appears to degrade at higher missing ratios (alpha = 0.5 or greater), especially under MNAR conditions (Figure 1 and Table 1). Table 1 indicates that the effectiveness of MissScore decreases significantly without variance reduction at lower noise levels (\sigma = 0.01), particularly for second-order scores, suggesting challenges in low-noise or high-missingness scenarios.

**Questions:**

• Pg. 2 typo: “We derives”

• Pg. 22 - What are the various synthetic data generation methods used to generate synthetic records in the TSTR?

• Could the authors comment on the observed instability of MissDAG and MissDiffAN in higher dimensions (d = 50), as seen in Figure 12?

• What might explain MissForest’s occasional competitiveness in certain conditions, in Figures 6-8?

---

> ### Author Response · Authors · 2024-11-20
> **Response to Reviewer bLh8**
>
> We thank the reviewer for their valuable insights and the time dedicated to reviewing our paper. We address each of the concerns and suggestions in detail below.
>
> > Changes Addressing Weaknesses
>
> We have incorporated your feedback on the **related work** concerning missing data and the **differences between MissScore and MissDiff** into Appendix Sections A and B. Additionally, we have outlined a few limitations of MissScore in the **Conclusion** section. All changes have been highlighted in blue for your reference. Thank you once again for your valuable suggestions.
>
> > Various synthetic data generation methods used to generate synthetic records in the TSTR.
>
> The baseline methods described in the **Baselines** paragraph of section 4 handle missing data by either imputing (filling in) or deleting it before training. These models are trained on a complete dataset (after imputation or deletion) and then generate synthetic data in the TSTR. In contrast, our approach trains directly on data with missing values and generates new synthetic data without needing to fill in or remove any missing entries beforehand.
>
> > The observed instability of MissDAG and MissDiffAN in higher dimensions (d = 50), as seen in Figure 12.
>
> Both MissDAG and MissDiffAN are expected to face challenges in high-dimensional settings (e.g. where d=50). Generally, MissDAG struggles with high-dimensional data, while MissDiffAN performs poorly with higher missing rates, as shown in Figures 7 through 11. MissDAG's limitations are due to the complex, computationally intensive nature of its EM approach. This iterative EM process, combined with rejection sampling, leads to scalability issues as the number of nodes increases, resulting in degraded performance and greater computational demands. MissDiffAN, on the other hand, requires substantial data for effective diffusion model training. However, biased imputation of missing values negatively impacts training, leading to unreliable performance. Surprisingly, MissDAG performs well in this scenario, consistent with results from MissOTM (Vo et al., 2024), where MissDAG also showed strong performance at a missing ratio with 50 nodes (see Figure 3). This may indicate that the specific characteristics of this dataset are favorable for MissDAG's approach.
>
> > MissForest’s occasional competitiveness in certain conditions, in Figures 6-8.
>
> MissForest’s occasional competitiveness in certain conditions, as seen in Figures 6-8, can be attributed to its unique strengths in handling complex data structures. In Figure 6, the task involves complete data, where DAGMA (Bello et al., 2022)—noted for its effectiveness in causal discovery within ANM—is applied, performs well. However, in Figures 7 and 8, where data imputation is required, MissForest emerges as a top performer due to its proven ability to deliver high-quality imputations. Studies such as MissOTM and MissDAG (Gao et al., 2022; Vo et al., 2024) consistently rank MissForest as one of the best imputers for causal discovery task. Its strengths lie in managing datasets with intricate interactions and non-linear relationships between variables of varying scales and types. Furthermore, MissForest performs effectively in high-dimensional datasets, even when the number of variables substantially exceeds the number of observations, consistently providing robust imputation results (Stekhoven & Buhlmann, 2012).
>
> References:
>
> Vy Vo, He Zhao, Trung Le, Edwin V Bonilla, and Dinh Phung. Optimal transport for structure learning under missing data. arXiv preprint arXiv:2402.15255, 2024.
>
> Kevin Bello, Bryon Aragam, and Pradeep Ravikumar. Dagma: Learning dags via m-matrices and a log-determinant acyclicity characterization. Advances in Neural Information Processing Systems,
> 35:8226–8239, 2022.
>
> Erdun Gao, Ignavier Ng, Mingming Gong, Li Shen, Wei Huang, Tongliang Liu, Kun Zhang, and Howard Bondell. Missdag: Causal discovery in the presence of missing data with continuous
> additive noise models. Advances in Neural Information Processing Systems, 35:5024–5038, 2022.
>
> Daniel J Stekhoven and Peter Buhlmann. Missforest—non-parametric missing value imputation for mixed-type data. Bioinformatics, 28(1):112–118, 2012.

---

> > ### Comment · Reviewer_bLh8 · 2024-11-21
> > **Acknowledgement of author response**
> >
> > I acknowledge that the authors have taken steps to improve the paper's presentation, including (1.) clarifying the differences between MissScore and MissDiff; (2) surveying related works on the "impute-then-predict" pipeline and multiple imputation; (3) acknowledging the limitations of MissScore; and (4) expanding on the strengths and occasional competitiveness of baseline methods like MissForest. I have increased the paper's presentation score and overall score accordingly.

---

### Official Review · Reviewer_opS8 · 2024-11-01

**Soundness:** 2
**Presentation:** 3
**Contribution:** 1
**Rating:** 3
**Confidence:** 4

**Summary:**

The paper presents a framework, MissScore, for estimating high-order scores in the presence of missing data, addressing a gap in data modeling with missing entries. The proposed method effectively integrates score-based models with missing data and provides theoretical guarantees under different missing mechanisms. Empirical results demonstrate that MissScore not only enhances the quality of data generation but also accelerates sampling, making it a competitive option for practical applications. However, I believe this manuscript, in its current form, requires further refinement.

**Strengths:**

MissScore integrates score-based models for data with missing entries, specifically focusing on higher-order scores. This provides a new perspective, enhancing data generation quality and sampling speed, which could benefit real-world applications. This framework also offers theoretical guarantees for different missing data mechanisms (e.g., MCAR and MAR), providing confidence in the approach's robustness and applicability under varying conditions. The empirical results validate the efficacy of MissScore, showing improvements over existing methods.

**Weaknesses:**

Limited Contribution over Existing Methods: this paper appears to primarily extend the work by Meng et al. (2021) by adapting it to scenarios with missing data. Experiments are limited to second-order scores (i.e., the Hessian of log density). Extending the analysis or at least discussing applications for higher-order scores (e.g., third-order scores) would enhance the practical relevance of the framework and could inspire broader applications. Additionally, the experiments rely exclusively on logistic regression for likelihood estimation, raising concerns about potential model misspecification and its impact on the results. The details can be found in the questions.

**Questions:**

1. In my view, this paper primarily extends the method proposed by Meng et al. (2021) to learn $n$-th order gradients of the log density (of perturbed data) in the presence of missing data. As is well known, under MCAR mechanism, it is often sufficient to use only complete-case data without applying imputation, and the resulting estimator remains consistent with the oracle estimator. (see Little and \& Rubin (2019)). It is unclear why the authors have chosen to focus on the MCAR mechanism specifically. A direct comparison between their proposed method and the approach by Meng et al. (2021), based solely on complete case data, would provide more meaningful insight. I hold a conservative opinion about the contribution
made by this paper.

2. In the conditions outlined for Theorems 6-7 under the MAR mechanism, there is no mention of the conditional propensity score function. Typically, with the IPW method under MAR, where the missing mechanism depends on observed variables, it is essential to assume that the conditional probability of missingness for each element lies strictly between 0 and 1, rather than imposing this requirement unconditionally. For example, what happens in Theorem 6 when $\mathbb{P}[m=0|x=x]\rightarrow 0$.

3. The experiments in this paper are limited to second-order scores (specifically, the Hessian of the log density). Could the authors suggest any practical applications where higher-order scores, such as third-order scores, might be useful?

4. In Section 3, it is unclear how to determine the tunable coefficient $\omega$.  At a minimum, the authors should provide guidelines for practitioners on how to select this parameter and perform a sensitivity analysis to evaluate its impact.

5. For missing data, it is common to consider the potential misspecification of the missing mechanism. However, in the experiments, the authors rely exclusively on logistic regression to estimate the likelihood. It would be valuable for the authors to conduct experiments that assess the impact of possible model misspecification.

References

[1] Meng, C., Song, Y., Li, W., \& Ermon, S. (2021). Estimating high order gradients of the data distribution by denoising. Advances in Neural Information Processing Systems, 34, 25359-25369.

[2] Little, R. J., \& Rubin, D. B. (2019). Statistical analysis with missing data (Vol. 793). John Wiley \& Sons.

---

> ### Author Response · Authors · 2024-11-20
> **Response to Reviewer opS8 (Part I)**
>
> We appreciate the reviewer’s thoughtful comments and the time spent on reviewing our paper. Below, we address each of the concerns and suggestions provided.
>
> > **Question 1:** Direct comparison between the proposed method and the approach by Meng et al. (2021), using only complete-case data and a conservative view on the paper’s contribution.
>
> In our work, we introduce a new approach to handling missing data with score matching. Our method addresses not only the MCAR case but also MAR case, ensuring that our estimator remains consistent with the oracle estimator. In MCAR, while using only complete-case data without imputation yields a consistent estimator comparable to the oracle, it performs worse when training a model (Ouyang et al., 2023; Kim et al., 2022; Gao et al., 2022; Vo et al., 2024), also as shown in the sampling experiments in Section 4. Additionally, at high missing rates, entire rows may be deleted, necessitating better methods for handling missing data in the MCAR scenario. Furthermore, our method supports k-th order estimation in the presence of missing data, which can be valuable for downstream tasks that require high-order score estimation.
>
> To provide meaningful insights, we compare our approach to the method by Meng et al. (2021), which is based on complete-case data only. In Table 1, we use the complete data (i.e., $\alpha=0.0$) to calculate the first- and second-order scores for comparison. As the missing ratio increases, the performance of our first- and second-order scores remains comparable to that of the complete data case, demonstrating robustness. We have also included the MAR results in Tables 3 and 4 of the updated appendix for further reference. Additionally, our focus was not limited to the MCAR mechansim; instead, we provide substantial evidence of our method's effectiveness across both MCAR and MAR scenarios through experiments in sampling (Section 4) and causal discovery (Section 5).
>
> > **Question 2:** Theorem 6 when $\mathbb{P}[m=0|x=x]\rightarrow0$.
>
> Thank you for pointing this out. Considering the case in Theorem 6 where $\mathbb{P}[\mathbf{m}=0|\mathbf{x}=\mathbf{x}] \to 0$, this has significant implications: the inverse probability weights, which are proportional to $1/\mathbb{P}[\mathbf{m}=0|\mathbf{x}=\mathbf{x}]$, become extremely large, leading to instability in the objective function. However, under the missing at random (MAR) assumption, the missing probability is generally treated as a fixed missing ratio in most studies, rather than approaching zero (Seaman & White, 2013; Vansteelandt et al., 2010; Zhang et al., 2023).
>
> > **Question 3:** Practical Applications with higher-order scores.
>
> For instance, third- and fourth-order scores can capture more nuanced properties of the denoising distribution $p(\mathbf{x}|\tilde{\mathbf{x}})$, such as estimating its skewness and kurtosis, thus providing deeper insights into its shape and behavior.
>
> > **Question 4:** Tunable coefficient $w$.
>
> The weight $\omega$ balances the contributions of the two loss components, $L_{\text{DSM}}(\theta)$ and $L_{\text{D2SM}}(\theta)$, in the overall objective function $L_{\text{joint}}(\theta)$. Selecting an appropriate $\omega$ is task-dependent and involves consideration of the relative importance of each score in the given context. For example, if the primary goal is to capture the broader structure of the data, as represented by the first-order scores, a smaller $\omega$ is preferred to place greater emphasis on the first-order term. However, if the task requires capturing finer-grained detail through higher-order relationships in the data, as is the case with second-order scores, a larger $\omega$ is advantageous and enhancing sensitivity to nuanced patterns. For consistency, $\omega$ is set to 10 across all experiments, as suggested by Meng et al. (2021).

---

> ### Author Response · Authors · 2024-11-20
> **Response to Reviewer opS8 (Part II)**
>
> > **Question 5:** Impact of possible model misspecification with logistic regression model under MAR.
>
> To illustrate the impact of potential model misspecification by the logistic regression model, we conduct an experiment comparing the ground truth $p$ with the estimated $\hat{p}$ from the logistic regression model on the synthetic dataset described in Section 3.2, with varying missing ratios. The results are provided in the additional discussions in Section B of the appendix, in Tables 3 and 4, in the updated manuscript. As you suggested, potential model misspecification in the logistic regression model does impact the estimation of missing probability. However, this effect remains within a reasonable range in the variance-reduced version.
>
> References:
>
> Stijn Vansteelandt, James Carpenter, and Michael G Kenward. Analysis of incomplete data using inverse probability weighting and doubly robust estimators. Methodology, 2010.
>
> Shaun R Seaman and Ian R White. Review of inverse probability weighting for dealing with missing data. Statistical methods in medical research, 22(3):278–295, 2013.
>
> Imke Mayer, Erik Sverdrup, Tobias Gauss, Jean-Denis Moyer, Stefan Wager, and Julie Josse. Doubly robust treatment effect estimation with missing attributes. The Annals of Applied Statistics, 14(3):1409–1431, 2020.
>
> Yuqian Zhang, Abhishek Chakrabortty, and Jelena Bradic. The decaying missing-at-random framework: Doubly robust causal inference with partially labeled data. arXiv preprint arXiv:2305.12789, 2023.
>
> Chenlin Meng, Yang Song, Wenzhe Li, and Stefano Ermon. Estimating high order gradients of the data distribution by denoising. Advances in Neural Information Processing Systems, 34:25359–25369, 2021.

---

### Official Review · Reviewer_xsPM · 2024-11-02

**Soundness:** 3
**Presentation:** 3
**Contribution:** 3
**Rating:** 6
**Confidence:** 3

**Summary:**

This paper presents a score estimation framework, namely MissScore, which aims to learn first and higher-order score functions in the presence of missing data. The key contribution of this paper is to provide theoretical results supporting the use of a score-based loss function for estimating higher-order scores in the presence of missing data. Based on the theory, the paper introduces an estimation methodology and demonstrates its effectiveness through a series of simulations and real data analyses.

**Strengths:**

The main strength of this paper lies in its reinforcement of the existing method with a more robust theoretical foundation. A similar loss function for estimating higher order score function was introduced earlier in Ouyang et al. but with no theoretical guarantees. This paper shows that (a variant of) the loss function is theoretically justified.  Based on theoretically justified, the authors built a method that outperforms other methods in the presence of the missing data.

**Weaknesses:**

One of this paper's key weaknesses is its presentation of theory. I have one major question, which I have listed below (in the Questions section). Other than that, the presentation needs to be involved. All the theorems can be viewed as an extended version of the score-matching theorems (both for the first order and the higher order score) in the presence of missing data. Hence, it is helpful to provide some remarks for readers with less technical background to clarify how these new theorems relate to the previous ones. Specifically, it would be useful to explain how you modify the earlier theorems to account for missing values and why these changes are intuitively reasonable.

**Questions:**

1. All the theorems (Theorem 1 - Theorem 7) are stated in terms of the joint distribution of $(x, \tilde x, m)$. However, we only have the distribution of $(x_{\rm obs}, \tilde x_{\rm obs}, m)$. Are all the theorems valid under this observed distribution?

2. What is the key difference between your method and the Missdiff of Ouyang et al.?

3. It is not immediate from the experiment section how your method performs compared to MissDiff. Can you present some real data analysis (say using Census data) to compare your method with MissDiff?

---

> ### Author Response · Authors · 2024-11-20
> **Response to Reviewer xsPM**
>
> We thank the reviewer for the comments, and we appreciate the time you spent on the paper. Below we address the concerns and comments that you have provided.
>
> > Provide additional details to ensure sufficient clarity
>
> Thank you for the suggestion! We will add more details to clarify how these new theorems relate to the previous score-matching theorems, both for the first-order and higher-order scores, in the presence of missing data, making it more accessible to readers with less technical background.
>
> > **Question 1:** All the theorems (Theorem 1 - Theorem 7) are stated in terms of the joint distribution of $(x, \tilde{x}, m)$. However, we only have the distribution of $(x_{\text{obs}}, \tilde{x}_{\text{obs}}, m)$. Are all the theorems valid under this observed distribution?
>
> No, the theorems do not hold if we rely solely on the observed data distribution. This limitation is primarily due to the score network, which is a neural network. However, element-wise multiplication with the mask naturally mitigates the impact of replacing missing values with zeros when computing the objective.
>
> > **Question 2:** Key difference between your method and the Missdiff of Ouyang et al.
>
> MissScore differs from MissDiff by focusing on learning high-order score with missing data, specifically leveraging the Hessian to enhance downstream performance. In contrast, MissDiff emphasizes unbiased data synthesis with a first-order diffusion-based approach. Additionally, MissScore introduces an oracle estimator under the MAR mechanism and introduce a novel causal discovery method that operates effectively in the presence of missing data.
>
> > **Question 3:** It is not immediate from the experiment section how your method performs compared to MissDiff. Can you present some real data analysis (say using Census data) to compare your method with MissDiff?
>
> MissDiff also conducted experiments on Census data, but since their code is not publicly available, we can only make indirect comparisons. Given that the diffusion model is, in some ways, equivalent to the DSM, MissDiff’s performance would be similar when applied to the Census data. However, our method incorporates second-order score information, which can provide better results. Therefore, our analysis primarily focuses on comparing first- and second-order models, where the first-order model aligns with the approach used in MissDiff. The results are presented in Section E.5 of the appendix in the updated manuscript.
>
> Our analysis consists of two parts: numerical and categorical data in the census dataset. For the numerical data, which includes variables such as age, final weight (fnlwgt), years of education completed (education-num), and hours worked per week, we examine the distribution of both the original data and the sampled data. This comparison is done using MissScore with first-order information via Langevin dynamics (equivalent to MissDiff) and MissScore with second-order information through Ozaki sampling. We observe that MissScore with second-order information captures finer details in the distribution. For instance, in the age range of 20-40, the fnlwgt range of 0-0.25, education-num range of 10-11, and hours per week range of 40-45, the second-order MissScore better approximates the shape of the original distribution.
>
> For the categorical data, we analyze the education variable and again find that the second-order MissScore aligns more closely with the original distribution. These results indicate that MissScore with second-order information captures finer-grained details in the sampled dataset, compared to MissDiff (MissScore with first-order information).

---

> > ### Comment · Reviewer_xsPM · 2024-11-25
> >
> > Thank you for your response. Although this explanation clarifies some of my concerns, my major concern was the applicability of the theories to the observed data, which is not addressed in the paper. Therefore, I am keeping my score.

---

### Official Review · Reviewer_9Bvm · 2024-11-02

**Soundness:** 2
**Presentation:** 3
**Contribution:** 2
**Rating:** 3
**Confidence:** 3

**Summary:**

The paper proposes a method called MissScore that estimates high-order scores with missing data, under both MCAR and MAR settings for second-order scores. The paper proves that the under appropriate assumptions, MissScore indeed estimates the score of the observed data. The paper compares MissScore with baselines on two applications of high-order score estimation.

**Strengths:**

The method and theory the paper introduces are novel to my knowledge, and address an important problem. The method outperforms alternatives in the experiments of the paper. The experiments are documented thoroughly.

**Weaknesses:**

The main weakness of the paper is the lack of realistic experiments. Only one real dataset is used, on only the sampling experiment. In the MAR experiments of the paper, the likelihood of each datapoint being observed is estimated with the same model that the missingness is assigned with. This is unlikely to be true in practice, so these should use different models in at least some experiments. Of course, even better would be using a real dataset with missing values, so there would be no need to artificially introduce missingness.

The paper claims that their results show that some baselines are introducing bias, since those baselines achieve worse results than MissScore. However, I don't think this is strong evidence of bias, since there could be other reasons for worse performance.

Minor points:
- Several spelling errors throughout the paper, for example fidelity, Langevin and Ozaki in Table 2.

**Questions:**

- The statements of Theorems 3-7 exclude the case where the probability of a missing value is 0, so all values are observed. What happens to the theory in this case? Do the theorems fail?
- The paper does not contain a theorem for $k$th order scores with $k > 2$ under MAR. Were you not able to prove this type of theorem? If so, what are the difficulties that make the general $k$ case harder in the MAR setting than the MCAR setting?
- What exactly is the fidelity metric measuring?
- How do you aggregate the utility metric values from the different predictive models?
- Is the number of iterations in Figure 2 the number of iterations for training the score models, or the number of iterations of Langevin dynamics?
- Why do the utility metrics for MissScore in Table 5 only decrease significantly with increasing missingness under MCAR, but not MAR or MNAR?

---

> ### Author Response · Authors · 2024-11-20
> **Response to Reviewer 9Bvm (Part I)**
>
> Thank you for your valuable comments and suggestions. We appreciate the time and effort you dedicated to reviewing our paper. Below, we address each of the concerns and feedback you provided.
>
> > The statements of Theorems 3-7 exclude the case where the probability of a missing value is 0, so all values are observed. What happens to the theory in this case? Do the theorems fail?
>
> Thank you for bringing up this important point. In Theorems 3-7, when all values are observed, the theorems remain valid. This corresponds to the original higher-order case, as discussed in Meng et al. (2021). We have incorporated the change in the revised version of our manuscript.
>
> > Theorem for k-th order scores with under MAR.
>
> Under the MAR assumption, the result will align with Theorem 5, except that the term $(\mathbf{1}-\mathbf{m})$ is incorporating the inverse probability weights. The proof will also follow the same structure as the proof of the original Theorem 5, with the adjustment for the inverse probability weights. I have revised and updated this result in Theorem 7, along with the corresponding proof. Thank you for bringing this to our attention.
>
> > Fidelity
>
> The *fidelity* metric measures how well synthetic data replicates the statistical properties of real data, specifically its distributions and inter-column relationships. In this work, we use [SDMetrics](https://docs.sdv.dev/sdmetrics/reports/quality-report/whats-included) to assess fidelity in two main areas: (1) Column Shapes: Assesses how closely the distribution in each synthetic column matches that of the real data. Using the Kolmogorov-Smirnov (KS) statistic, which compares CDFs, SDMetrics calculates the KSComplement (1 - KS statistic), where a higher score indicates a closer match; (2) Column Pair Trends: Evaluates if the relationships (e.g., correlation) between column pairs in synthetic data align with those in the real data. The overall fidelity score averages these two components into a single metric of similarity, which we use to compare data quality, as in Ouyang et al. (2023).
>
> > How do you aggregate the utility metric values from the different predictive models?
>
> The utility metrics—Accuracy, AUROC, and weighted-F1—are reported separately for each predictive model without aggregation. In Table 2 of the main text, utility is represented by Accuracy, while additional evaluations, including AUROC and weighted-F1, are provided in Appendix Tables 6 and 7.
>
> > Is the number of iterations in Figure 2 the number of iterations for training the score models, or the number of iterations of Langevin dynamics?
>
> Apologies for the confusion—the number of iterations in Figure 2 refers to the number of iterations of Langevin dynamics. We have made updates in the revised version of our manuscript.
>
> > Why do the utility metrics for MissScore in Table 5 only decrease significantly with increasing missingness under MCAR, but not MAR or MNAR?
>
> Thank you for bringing up this important point. The observed results can be explained by the nature of the missingness mechanisms. Under MCAR, data is missing independently of both observed and unobserved values. As the missing rate increases, information is lost randomly across samples and features, leading to a consistent degradation in model performance. In contrast, MAR and MNAR involve dependencies that allow the model to better handle missing data. Under MAR, missingness depends on observed data, enabling the model to leverage correlations between observed variables and missingness patterns. With MNAR, missingness is related to unobserved data (e.g., missing values correlate with the missing data points themselves). Although MAR and MNAR generally present challenges for imputation, some methods can incorporate assumptions about these mechanisms to manage missing data more effectively (Van buuren & Groothuis-Oudshoorn, 2011; Gondara & Wang, 2018; Gain & Shpitser, 2018; Wang et al., 2021; Ouyang et al., 2023). Depending on the strength of these dependencies, the model may be able to sample new data more accurately from the learned model.

---

> ### Author Response · Authors · 2024-11-20
> **Response to Reviewer 9Bvm (Part II)**
>
> > The paper claims that their results show that some baselines are introducing bias, since those baselines achieve worse results than MissScore. However, I don't think this is strong evidence of bias, since there could be other reasons for worse performance.
>
> Yes, however, pinpointing a single reason for the poorer performance of these imputation methods is challenging. One contributing factor might be that some imputation methods don’t fully account for the uncertainty inherent in missing data, which can impact performance. Regarding bias, as Ouyang et al. (2023) shows, an “impute-then-generate” approach may introduce bias by potentially reducing generation quality, which can affect downstream tasks. Additionally, as Kyono et al. (2021) note, conditioning on missingness indicators during imputation may create spurious dependencies. In contrast, MissScore learns distributions directly from the data without relying on imputation, may help alleviate these issues. However, for some tasks, methods like MissDAG and MissOTM—which use directed acyclic graph (DAG) structures as inductive biases—could achieve better results, especially in causal discovery tasks where structured dependencies offer an advantage.
>
>
> References:
>
> Chenlin Meng, Yang Song, Wenzhe Li, and Stefano Ermon. Estimating high order gradients of the data distribution by denoising. Advances in Neural Information Processing Systems, 34:25359–25369, 2021.
>
> Yidong Ouyang, Liyan Xie, Chongxuan Li, and Guang Cheng. Missdiff: Training diffusion models on tabular data with missing values. arXiv preprint arXiv:2307.00467, 2023.
>
> Stef Van Buuren and Karin Groothuis-Oudshoorn. mice: Multivariate imputation by chained equations in r. Journal of statistical software, 45:1–67, 2011.
>
> Lovedeep Gondara and Ke Wang. Mida: Multiple imputation using denoising autoencoders. In
> Advances in Knowledge Discovery and Data Mining: 22nd Pacific-Asia Conference, PAKDD
> 2018, Melbourne, VIC, Australia, June 3-6, 2018, Proceedings, Part III 22, pp. 260–272. Springer, 2018.
>
> Alexander Gain and Ilya Shpitser. Structure learning under missing data. In International conference on probabilistic graphical models, pp. 121–132. Pmlr, 2018.
>
> Zhenhua Wang, Olanrewaju Akande, Jason Poulos, and Fan Li. Are deep learning models superior for missing data imputation in large surveys? evidence from an empirical comparison. arXiv preprint arXiv:2103.09316, 2021.
>
> Trent Kyono, Yao Zhang, Alexis Bellot, and Mihaela van der Schaar. Miracle: Causally-aware imputation via learning missing data mechanisms. Advances in Neural Information Processing Systems, 34:23806–23817, 2021.

---

> ### Comment · Reviewer_9Bvm · 2024-11-21
>
> Thank you for the reply. You addressed my minor concerns and comments, but I did not see anything about my comment on the lack of realistic experiments. Since that is the main weakness of the paper in my opinion, I'm keeping my score for now.

---

### Meta-Review · Area_Chair_bwrU · 2024-12-20

**Metareview:**

The paper presents a method for higher order score estimation in the presence of missing data.

In terms of strengths, the reviewers find the paper to address an important novel problem and provide a strong theoretical framework. The presentation is clear.

In terms of weaknesses that remain after the rebuttal and revision the reviewers are concerned about lack of realistic experiments and experiments on scores with order >2, as well as limited contribution.

Considering all the reviews, it seems clear that the submission does not reach the bar of acceptance to ICLR due to weaknesses stated above. None of the reviewers strongly support the submission, and at least one clearly recommends rejection even after the rebuttal.

**Additional Comments On Reviewer Discussion:**

The authors addressed many of the points raised by the reviewers but were unable to answer some fundamental concerns. There was no further discussion because the decision was clear.

---

### Decision · Program_Chairs · 2025-01-22

Reject